# The Human Pathogen *Mycobacterium tuberculosis* and the Fish Pathogen *Mycobacterium marinum* Trigger a Core Set of Late Innate Immune Response Genes in Zebrafish Larvae

**DOI:** 10.3390/biology13090688

**Published:** 2024-09-03

**Authors:** Ron P. Dirks, Anita Ordas, Susanne Jong-Raadsen, Sebastiaan A. Brittijn, Mariëlle C. Haks, Christiaan V. Henkel, Katarina Oravcova, Peter I. Racz, Nynke Tuinhof-Koelma, Malgorzata I. Korzeniowska nee Wiweger, Stephen H. Gillespie, Annemarie H. Meijer, Tom H. M. Ottenhoff, Hans J. Jansen, Herman P. Spaink

**Affiliations:** 1ZF-Screens B.V., J.H. Oortweg 19, 2333 CH Leiden, The Netherlandsb.brittijn@gmail.com (S.A.B.); christiaan.henkel@nmbu.no (C.V.H.); petiracz@yahoo.com (P.I.R.); koelmany@gmail.com (N.T.-K.); mwiweger@gmail.com (M.I.K.n.W.); jansen@futuregenomics.tech (H.J.J.); 2Department of Infectious Diseases, Leiden University Medical Center, 2300 RC Leiden, The Netherlandst.h.m.ottenhoff@lumc.nl (T.H.M.O.); 3School of Biodiversity, One Health and Veterinary Medicine, University of Glasgow, Jarrett Building, Glasgow G61 1QH, UK; katarina.oravcova@glasgow.ac.uk; 4Medical and Biological Sciences Building, University of St Andrews, North Haugh, St Andrews, Fife KY16 9TF, UK; shg3@st-andrews.ac.uk; 5Institute of Biology, Leiden University, 2333 CC Leiden, The Netherlands; a.h.meijer@biology.leidenuniv.nl

**Keywords:** bacterial robotic injection, host infection transcriptome, Illumina RNAseq, automated infection, innate immune response

## Abstract

**Simple Summary:**

This study investigates zebrafish larvae as a model for *Mycobacterium tuberculosis* infection, a major cause of tuberculosis in humans. Despite not being natural hosts, zebrafish larvae are successfully infected with *M. tuberculosis*, showing propagation for up to 9 days post-injection using a robotic system for efficiency. Fluorescence microscopy confirms microbial aggregates carrying labeled *M. tuberculosis*, mirroring the infection by *M. marinum*, a related surrogate model. Transcriptome analyses shows that *M. tuberculosis* triggers a specific transcriptional immune response in infected larvae, resembling the response to *M. marinum*. The study demonstrates the persistence of *M. tuberculosis* in zebrafish larvae for at least a week post-infection. It supports the use of the *M. marinum* infection model as a surrogate for tuberculosis for comparing key immune response genes induced by *M. tuberculosis*. Additionally, the research compares the zebrafish model’s response to *M. tuberculosis* with human macrophage responses, revealing shared and unique gene expression patterns. This work contributes to efficient preclinical tools for tuberculosis research, underscoring the potential of zebrafish as a model host for discovering diagnostic markers of *M. tuberculosis* infections and for confirming the effects of potential drugs identified using the *M. marinum* infection model.

**Abstract:**

Zebrafish is a natural host of various *Mycobacterium* species and a surrogate model organism for tuberculosis research. *Mycobacterium marinum* is evolutionarily one of the closest non-tuberculous species related to *M. tuberculosis* and shares the majority of virulence genes. Although zebrafish is not a natural host of the human pathogen, we have previously demonstrated successful robotic infection of zebrafish embryos with *M. tuberculosis* and performed drug treatment of the infected larvae. In the present study, we examined for how long *M. tuberculosis* can be propagated in zebrafish larvae and tested a time series of infected larvae to study the transcriptional response via Illumina RNA deep sequencing (RNAseq). Bacterial aggregates carrying fluorescently labeled *M. tuberculosis* could be detected up to 9 days post-infection. The infected larvae showed a clear and specific transcriptional immune response with a high similarity to the inflammatory response of zebrafish larvae infected with the surrogate species *M. marinum*. We conclude that *M. tuberculosis* can be propagated in zebrafish larvae for at least one week after infection and provide further evidence that *M. marinum* is a good surrogate model for *M. tuberculosis*. The generated extensive transcriptome data sets will be of great use to add translational value to zebrafish as a model for infection of tuberculosis using the *M. marinum* infection system. In addition, we identify new marker genes such as *dusp8* and CD180 that are induced by *M. tuberculosis* infection in zebrafish and in human macrophages at later stages of infection that can be further investigated.

## 1. Introduction

Tuberculosis (TB) causes more than a million casualties every year, and as a result of the emergence of multidrug-resistance (MDR-TB) and extensive drug-resistance (XDR-TB), there is an urgent need for new and more effective TB therapies [1,2]. Preclinical tools play a crucial role in the development of new drug therapies, and the better these tools are, the higher the chances of success in subsequent clinical trials. Preclinical TB research relies on various in vitro and in vivo model systems [3,4]. The majority of in vitro models are based on cultured or primary human macrophages, which are the natural host cells of *Mycobacterium tuberculosis*, the causative agent of the disease [5]. Some large mammals, such as cows and goats, are natural hosts of tuberculosis, but for obvious reasons (ethics, size, costs) they are hardly being used in the drug development process [6,7,8,9]. Instead, mice, rabbits and guinea pigs are the most commonly used preclinical TB models, although they are not natural hosts of *M. tuberculosis* [10,11].

Several fish species, including the popular laboratory model organism zebrafish (*Danio rerio*), are a natural host of *M. marinum,* which causes fish tuberculosis [5,12]. *M. marinum* is the evolutionarily closest non-tuberculous relative of *M. tuberculosis*, with a 50% larger genome (6.64 Mbp versus 4.41 Mb) and ~1500 more genes (5503 versus 4008) than *M. tuberculosis* (Table 1; [5]). *M. marinum* is also pathogenic to humans, but since it is not viable at 37 °C, its disease symptoms are usually limited to the extremities, where it can cause a skin condition known as “swimming pool or fish tank granuloma” [13,14,15].

For more than a decade, the zebrafish and *M. marinum* host–pathogen system has been used as a surrogate model for human TB [16]. As an experimental laboratory model, *M. marinum* has some clear advantages compared with *M. tuberculosis*, including a much shorter doubling time (~4 h instead of ~20 h) and less stringent safety restrictions (Biosafety level 2 (BSL-2) instead of BSL-3) [5]. Although zebrafish have an optimum temperature of approximately 28 °C, they can tolerate temperatures from 8 °C to 42 °C [17,18,19]. Thus, in principle, it should be possible to grow adult zebrafish at a temperature of 37 °C, which is the optimum temperature of *M. tuberculosis*; however, the feasibility of using zebrafish as a host for *M. tuberculosis* has hardly been studied.

Earlier, we described a robot for the automated intra-yolk injection of zebrafish embryos as part of the ongoing efforts to develop the zebrafish larvae system into a high-throughput in vivo vertebrate drug screening platform for infectious diseases and cancer [20,21]. We also described the robotic injection of *M. tuberculosis* into zebrafish embryos, showing, as an additional advantage, the considerably reduced health risk and hands-on time for scientists. Viable *M. tuberculosis* could be recovered from zebrafish larvae up to 6 days post-infection (6 dpi) [20,21]. For long-term drug treatment studies, it would be desirable to culture *M. tuberculosis*-infected larvae for prolonged periods of time. In the present study, we examined whether *M. tuberculosis* would survive for more than a week in zebrafish larvae. We also measured the effect of *M. tuberculosis* infection on the larval transcriptome via Illumina RNAseq and compared it with the effect of *M. marinum* infection to validate known markers and investigate the specificity of the responses to these mycobacteria. Furthermore, a comparison was made with the transcriptome profile of human macrophages in response to *M. tuberculosis* infection. We conclude that *M. tuberculosis* can survive in zebrafish larvae for at least one week after infection and provide further evidence that *M. marinum* is a good surrogate model for *M. tuberculosis*. Our RNAseq data set of zebrafish larvae infected with *M. tuberculosis* from 3 to 8 days post-infection is valuable for comparisons with current and future transcriptome studies of mycobacterial infectious disease. In addition we identify new marker genes such as *dusp8* and *CD180* that are induced by *M. tuberculosis* infection in zebrafish and in human macrophages at later stages of infection that can be further investigated.

## 2. Materials and Methods

### 2.1. Bacterial Culture

Reference strain *M. tuberculosis* H37Rv (Mtb; obtained from the Kaufman lab, MPI, Berlin, Germany) and its DsRed-labeled derivative strain (Mtb^DsRed^) were grown in Middlebrook 7H9 medium (Becton, Dickinson U.K. Limited, Wokingham, UK) at 37 °C in a BSL-3 laboratory.

### 2.2. Zebrafish Experiments Ethics Statement

All zebrafish larvae studies were ethically reviewed and approved by the local Animal Experiments Committee (DEC-application Nr. 12024u) and carried out in accordance with European Directive 2010/63/EU. Wild-type zebrafish stocks of the AB/TL line were handled in compliance with the local animal welfare regulations and maintained according to standard protocols (zfin.org). Zebrafish stock fish were maintained at the zebrafish facility of Leiden University.

### 2.3. Zebrafish Embryo Injections and Larvae Culturing

Initially, considerable effort was invested in determining the optimal growth conditions of *M. tuberculosis*-infected zebrafish larvae, also taking into account the strict safety restrictions associated with working with BSL-3 pathogens. Despite the broad temperature tolerance of adult zebrafish, larval mortality rapidly increases above 32 °C and is almost 100% at 36 °C [22]). Therefore, 34 °C was chosen as a compromise between increased larval death and decreased metabolic activity of *M. tuberculosis*.

Zebrafish embryonic development rate is temperature-dependent and at 34 °C embryos reach the feeding larval stage ~30% faster than at 28 °C. Oxygen tension is reduced at higher temperature; however, providing water aeration is not advised to avoid aerosol formation and minimize the risk of *M. tuberculosis* spread during the experimental work under BSL-3 regulations. As water becomes more polluted from zebrafish food at higher temperatures, leading to increased larval mortality, *M. tuberculosis*-infected feeding larvae (>4 dpi) were cultured in T75 tissue culture flasks with limited water (75 mL). Every day, 50% of the water was replaced with fresh water. Survival scoring of larvae was based on presence of motility and heartbeat of the larvae.

Synchronized zebrafish AB/TL strain embryos were obtained using an iSPAWN (Tecniplast, Buguggiate, Italy) or collected from manually arranged family crosses and transferred to egg water (60 μg/mL Instant Ocean sea salts, Sera Marin). Early embryos (16-128-cell stage) were infected via robotic intra-yolk injection with 500 colony-forming units (CFU) *M. tuberculosis* H37Rv (wild-type for RNA isolation and CFU counts) and *M. tuberculosis*-DsRed (for imaging and CFU counts), as previously described [20,23]. Briefly, the synchronized embryos were divided over a 1024-well agarose grid and injected with 1 nL bacterial suspension in 2% polyvinylpyrrolidone (PVP) in phosphate-buffered saline (PBS) using a robotic injector (Life Science Methods, Leiden, The Netherlands) [20] equipped with a custom-designed injection needle with 10 μm inner tip diameter (Qvotek, Mississauga, Canada). As a control, a group of embryos were injected with the 2% PVP-PBS solution [20]. Upon injection, the embryos were incubated at 34 °C in Petri dishes (50 embryos per dish) until four days post-fertilization (dpf). On day 4, larvae were transferred to T75 cell culture flasks (Greiner Bio-One Cat. #658175) each with 20 larvae in 75 mL egg water (flasks laid horizontally). Half of the egg water was refreshed daily. Larvae from 4 dpf were fed twice a day with powdered dry food (DuplaRin M, Gelsdorf, Germany). The light/dark regime in the incubator was 14:10 h.

### 2.4. Fluorescence Microscopy

For microscopy, 0 and 6–9 dpf *M. tuberculosis*-DsRed injected and control larvae (*n* = 7–20) were fixed in 4% paraformaldehyde in PBS overnight at 4 °C. To remove any extracellular bacteria, amikacin treatment for 1 h at room temperature in 1:1000 in PBS was applied after fixation [24]. After the treatment, the larvae were washed three times and batches of 30 larvae per group were maintained in PBS at 4 °C for imaging. Fluorescence in larvae was observed using a Leica MZ16 FA fluorescence stereomicroscope and a Leica TCS SPE laser-scanning confocal microscope (Leica Microsystems, Amsterdam, The Netherlands).

### 2.5. CFU Counting

For establishing viable bacterial counts as CFU in infected larvae, larvae were subjected to BBL™ MycoPrep™ treatment (BD Biosciences Cat. # 240862) to digest/decontaminate samples suspected to contain mycobacteria and plated out on Middlebrook 7H10 agar supplemented with Oleic-Albumin-Dextrose-Catalase (OADC) growth supplement (both from Becton, Dickinson U.K. Limited, Wokingham, UK) [25,26].

At 0 dpf and 6–8 dpf, larvae (*n* = 3–5 in triplicates) were collected from the *M. tuberculosis*-DsRed and PVP injected groups and 100 μL 5% sodium dodecyl sulfate (SDS) in PBS along with a sterile 5 mm stainless steel bead was added to a 2 mL reaction vial. After homogenization by vortexing, 100 μL MycoPrep™ was added to the samples and incubated shaking for 10 min. To neutralize the reaction, 1.8 mL PBS was pipetted to the tube. All suspensions were diluted, plated in duplicate on Middlebrook 7H10 agar Supplemented with OADC (without any antibiotic addition) and incubated at 37 °C. After 26 days, colonies were counted.

### 2.6. RNA Isolation, MBL Assay and Quantitative RT-PCR

Embryos/larvae injected with wild-type *M. tuberculosis* or PVP alone were washed three times with egg water and harvested at 0–9 dpf for RNA isolation. The number of replicates and larvae used per experiment are shown in Figure 1. At least duplicates or triplicates were used for transcriptome profiling. These samples were the same as used for the Q-PCR analysis. Subsequently, 750 μL Qiazol was added and samples were kept at −20 °C until further processing. Total RNA was isolated from larvae using the Qiagen miRNeasy mini kit according to the manufacturer’s instructions (Qiagen, Venlo, The Netherlands). Quality and integrity of the RNA was checked on an Agilent Bioanalyzer 2100 total RNA Nano series II chip (Agilent, Amstelveen, The Netherlands).

For CFU determination using the molecular bacterial load (MBL) assay, RNA samples from the following time points were selected: 0, 1, 2, 3, 4, 5 and 9 dpf. Internal control (IC) RNA and RT-qPCR primers and probes were provided by University of St. Andrews. The amplification of *M. tuberculosis* 16S rRNA (ROX) and IC (JOE) was measured according to method previously described [27]. The samples were normalized to IC recovery, and from that number the amount of CFU per embryo was calculated by multiplying the volume of Qiazol in mL used for the RNA isolation by the amount of CFU per mL and then dividing this by the number of embryos per replicate. Raw data and the resulting derivative quantitations and standard deviations are provided in Appendix A.

For quantitative RT-PCR (Q-PCR) analysis of matrix metallopeptidases 9 (*mmp9*) and interleukin-1 beta (*il1b*) mRNA levels, RNA samples from the following time points were selected: 0, 1, 2, 3, 4, 5, 6, 8 and 9 dpi. We followed the protocol of Jiménez-Amilburu et al. [28] that is briefly described here. All samples were treated with RQ1 DNAse (Promega, Southampton, UK) to remove any residual genomic DNA. Total RNA was reverse transcribed into cDNA using IScript Reverse Transcriptase (Bio-Rad Laboratories, Hercules, CA, USA). Conventional PCR was performed using DreamTaq Green DNA Polymerase (Thermo Scientific, Rockford, IL, USA), following manufacturer’s indications. qPCR analysis was performed using iQ SYBR Green supermix (Bio-Rad Laboratories B.V.). The reactions were run in an iCycler Thermal Cycler (Bio-Rad Laboratories B.V.) using the following protocol: 3 min at 95 °C, followed by 40 steps of 15 s denaturation at 95 °C, 30 s at the corresponding annealing temperatures and a final melting curve of 81 steps from 55 °C to 95 °C (0.5 °C increments every 10 s) [28].

### 2.7. Statistical Analysis

Statistical analysis of the Q-PCR data was performed using the online analysis web site https://www.statskingdom.com/170median_mann_whitney.html (accessed on 17 February 2024). The Mann–Whitney U test (two-tailed) (Wilcoxon Rank Sum) was used considering that the data scored unfavorable for normality. The raw data and statistical analyses are presented in Appendix A. For the survival analysis, we performed Mantel–Cox test using GraphPad Prism version 10: https://www.graphpad.com/guides/prism/latest/user-guide/survival_table.htm?q=log-rank+test (accessed on 3 March 2024).

### 2.8. Illumina RNAseq Analysis

Illumina RNAseq libraries were prepared from 2 μg total RNA using the Illumina TruSeq™ RNA Sample Prep Kit v2 according to the manufacturer’s instructions (Illumina Inc., San Diego, CA, USA). All RNAseq libraries (150–750 bp inserts) were sequenced on an Illumina HiSeq2500 sequencer as 1 × 50 bp single read runs according to the manufacturer’s protocol. Image analysis and base calling were conducted by the Illumina pipeline. The raw RNAseq data have been submitted to the NCBI GEO database as project GSE252417, Acc. Nrs. GSM8001074-GSM8001105.

### 2.9. Illumina Data Processing

For transcriptome profiling, the genes expressed with significant change (*p*-value < 0.05, fold change >2 and fold change <−2) were selected. Data processing was performed as described previously [23]. Lists of significantly changed genes were analyzed using a web-based annotation platform KOBAS 2.0 (KEGG Orthology Based Annotation System) that classifies genes into KEGG pathways and Gene Ontology terms (http://kobas.cbi.pku.edu.cn/home.do, accessed on 15 December 2022) [29]. KEGG pathways and GO terms with *p*-value < 0.05 were considered significant and used for further analysis. For comparison, *M. marinum* data were extracted from Benard et al. [30], while the human macrophage data infected with Mtb was obtained from Vrieling et al. [31].

## 3. Results

### 3.1. Long-Term Survival of M. tuberculosis in Zebrafish Larvae

Three different strategies were followed to monitor the presence of *M. tuberculosis* in infected zebrafish embryos/larvae: (1) fluorescence microscopy of DsRed-labeled mycobacteria to monitor infection, (2) viable mycobacteria counts (in CFU) from larval lysates treated with MycoPrep^TM^ and plated on agar plates and (3) RNA analysis of embryos and larvae harvested in Qiazol.

Early zebrafish embryos were robotically injected into the yolk with carrier control (PVP), wild-type Mtb or DsRed-labeled Mtb (Mtb^DsRed^) and subsequently incubated at 34 °C. The groups of embryos/larvae were harvested at multiple time points and processed according to the experimental design depicted in Figure 1. At 1 dpf, embryonic survival in the Mtb group was 50%, whereas 30% of the embryos survived in the Mtb^DsRed^, PVP and uninjected groups (Figure 2). At 4 dpf, 24% of the Mtb, 12% of the Mtb^DsRed^, 25% of the PVP and 26% of the uninjected larvae were still alive. Larval survival at 7 dpf was about 11% in the Mtb group, 5% in the Mtb^DsRed^ group and 20% in the PVP and uninjected groups. At 8 dpf, 3% of the Mtb and Mtb^DsRed^ groups, 5% of the PVP group and 10% of the uninjected group was still alive. The last viable larvae could be collected at 9 dpf, when 1% of the three injected groups and 7% of the uninjected control group was still alive. Since the experiment was only performed in duplicate for the early time points, the power of the statistical analysis is limited (Appendix A). However, in general, the data show no relevant difference in survival in the presence or absence of injected Mtb bacteria since the effect of the growth temperature at 34 °C on the survival curves in the absence of infection is so strong that it makes differences between individual strains insignificant in comparison. In contrast, the robotic injection of *M. marinum* at the growth temperature of 28 °C does not affect survival when the larvae are treated with antibiotics (Appendix A). This shows that our robotic injection method and setup does not influence survival at 28 °C up to 7 dpi. Therefore, it is clear that after 7 dpi, the number of surviving larvae at the growth temperature of 34 °C becomes limiting for the analysis of responses to infection with *M. tuberculosis*, in contrast to studies with *M. marinum* at 28 °C.

To check if the embryos were properly injected with the bacteria, we first performed fluorescence stereomicroscopy on embryos of the Mtb^DsRed^ groups that were fixated right after injection. Figure 3 shows a representative image of an infected embryo (0 dpi), demonstrating that fluorescent bacteria were successfully injected into the yolk. At 6 dpi half of the fixated larvae (nine out of seventeen) showed granuloma-like aggregates spread throughout the body and the rest had signal only in the yolk. At 8 dpi, four out of seven collected larvae had detectable fluorescent signal with few granulomas in the body, and at 9 dpi, only one out of seven imaged larvae showed detectable fluorescence spread throughout the body (Figure 3).

As another means to measure the presence of viable *M. tuberculosis*, larvae from each injection group (3 × *n* = 5) were treated with MycoPrep and plated on agar plates at 0, 6 and 8 dpi. After 26 days of culturing only lysates derived from 0 dpf embryos of the Mtb^DsRed^ group resulted in bacterial colonies (68, 78 and 220 CFU per plate), whereas the other plates were either overgrown with other bacteria or did not show bacterial growth at all.

Since MycoPrep treatment of samples often results in poor recovery of *M. tuberculosis*, we also used the molecular bacterial load (MBL) assay to test for the presence of mycobacterial RNA in the zebrafish larvae samples. Embryos were injected with either *M. tuberculosis* H37Rv (no plasmid) in PVP or with PVP alone (negative control). At 0, 1, 2, 3, 4, 5, 6, 8 and 9 dpi, three to six groups of 5–10 pooled PVP- or *M. tuberculosis*-injected embryos/larvae were harvested for RNA isolation. RNA samples of 0, 1, 2, 3, 4, 5 and 9 dpi were selected for CFU determination using the MBL assay. An average CFU value of ~350 CFU was detectable with a maximum of ~600 CFU at 2 dpi and a minimum of ~100 CFU at 9 dpi (Figure 4 and Appendix A). Since the biological experiment was only performed once, no statistical analysis was possible, and the results are only qualitative. However, considering the low standard deviations for a total of five larvae per time point and the sensitivity of the MBL method, we can conclude that *M. tuberculosis* can survive up to 9 days after the infection of zebrafish embryos, but at very low numbers. This is not surprising, considering that a 9 days period of growth is still very short for *M. tuberculosis* growing at a suboptimal temperature in a hostile environment. However, these low numbers will make any accurate quantitative analysis using follow up MBL experiments very difficult, if not impossible. Considering the low numbers of larvae available and the conclusive evidence from the microscopy results, we therefore have not replicated the experiment.

### 3.2. Transcriptional Profiling of Zebrafish Larvae Infected with M. tuberculosis

The expression of the proinflammatory marker genes *il1b* and *mmp9* is strongly induced in zebrafish larvae shortly after infection with *M. marinum* [30]. A similar increase in expression of the *il1b* and *mmp9* genes was observed via qPCR analysis of *M. tuberculosis*-infected larvae (Figure 5). *Il1b* expression was increased at 2, 3, 4, 5, 6 and 8 dpi with the highest increase (16-fold) found at 5 dpi. *Mmp9* expression was increased at 3, 4, 5, 6 and 8 dpi and also peaked at 5 dpi (18-fold increase). For statistical analysis of the qPCR results, samples from 1 + 2 dpi, 3 + 4 dpi, 5 + 6 dpi, 8 + 9 dpi were analyzed together to have a sufficiently large sample number to be able to perform a Mann–Whitney U test. Significantly different expression levels between PVP control and Mtb injected larvae are indicated by an asterisk in Appendix A. In conclusion, the qPCR analysis showed that there was a significant innate immune response to the infection with *M. tuberculosis* in the combined 5- and 6-day time points. The raw data (Appendix A) show that there is very low expression of *il1b* and *mmp9* in the absence of infection in all cases, as expected from the published literature for these markers in zebrafish larvae [30]. Therefore, although here is a high standard deviation as is usual for these markers in zebrafish infection assays [30], from a qualitative standpoint, there is a clear inflammatory response still at 9 days post-infection with *M. tuberculosis*.

To study the late-phase transcriptional host response to *M. tuberculosis* infection in more detail, we proceeded with Illumina deep transcriptome sequencing of the following selection of biological replicates of the time series: 3 dpi (*n* = 3), 4 dpi (*n* = 3), 5 dpi (*n* = 3), 6 dpi (*n* = 3) and 8 dpi (*n* = 2). Each biological replicate consisted of 5–10 pooled larvae (Figure 1). Statistical DESeq analysis was used to identify genes that were differentially expressed in *M. tuberculosis*-infected larvae compared with PVP-injected larvae (*p* < 0.05; fold change > 2.0 and fold change < −2) (Figure 6A). At 3 dpi, 556 genes showed higher and 43 genes showed lower expression in Mtb-infected compared with PVP-treated larvae. At 4, 5 and 6 dpi, the number of genes with increased expression remained constant at 351, 394 and 374, respectively, whereas the number of genes with reduced expression steadily increased from 132 (4 dpi) to 159 (5 dpi) and 381 (6 dpi). At 8 dpi, the number of differentially expressed genes showed a massive increase: 1345 genes showed higher expression and 861 genes showed lower expression in Mtb-infected versus mock-treated larvae. Pair-wise comparisons of consecutive time points (Figure 6B) showed that, between most time points, the overlap of genes with increased expression was less than 25% (80–90 genes), except for around 45% overlap (171 genes) between 5 and 6 dpi. The overlap of genes with reduced expression was even less: 4–12% (5–19 genes) between most time points and only about 1% (4 genes) between 6 dpi and 8 dpi. Despite the relatively low overlap of up-regulated genes between two consecutive time points, a core set of 23 genes was always up-regulated at all five examined time points; however, none of the down-regulated genes overlapped at all five time points (overlap in the center of Figure 6C). The core set of 23 up-regulated genes included several known immune response markers, such as *mmp9*, matrix metallopeptidase 13a (*mmp13a*), chemokine receptor 12b2 (*ccr12b*) and DNA damage-regulated autophagy modulator 1 (*dram1*) (Table 2).

### 3.3. Effect of Increased Temperature on Transcriptome Profile of Zebrafish Larvae

Larval mortality was significant at temperatures above 32 °C in the presence or absence of infection, which is in line with a recent study [22]. To obtain an impression of the effect of the increased temperature on the transcriptome profile of uninfected zebrafish larvae, we compared the 3, 4 and 5 dpi PVP-injected larvae of the Mtb experiment (34 °C) with the developmental stage-matched 3, 4, 5 dpi (4, 5, and 6 dpf) PVP-infected larvae of the *M. marinum* experiment (28.5 °C). A massive number of genes was differentially expressed in the 34 °C and 28.5 °C groups at different time points, namely 7705 at 3 dpi (3481 increased and 4224 decreased), 8635 at 4 dpi (3998 increased and 4637 decreased) and 9363 at 5 dpi (4696 increased and 4667 decreased), corresponding with, respectively, 24%, 27% and 29% of all annotated genes. Similar numbers were found when the 3, 4 and 5 dpi Mtb-infected larvae (34 °C) were compared with the 3, 4 and 5 dpi *M. marinum*-infected larvae (28.5 °C), namely 9979 at 3 dpi (4685 increased and 5294 decreased), 9239 at 4 dpi (3795 increased and 5444 decreased) and 9547 at 5 dpi (4651 increased and 4896 decreased), corresponding with, respectively, 31%, 29% and 30% of all annotated genes. Although part of these differences was most likely caused by the non-synchronous developmental stages of the two groups, 38–50% of the differentially expressed genes overlap between the three consecutive days (Appendix A) and are therefore expected to be caused by the increased temperature. Even if all 34 °C groups (PVP and Mtb) are compared with all 28.5 °C groups (PVP and *M. marinum*), the majority of differentially expressed genes still overlaps (Appendix A), further supporting that these differences are mainly the result of increased temperature. Indeed, the 1819 up-regulated genes include all major genes encoding heat stress-related proteins, such as Hsp40, Hsp70 and Hsp90. GO analysis of this list shows significant enrichment in various KEGG pathway related to cytoskeleton and metabolism (Appendix A).

### 3.4. Comparison of Host Transcriptional Response between Zebrafish Larvae Infected with M. tuberculosis and Larvae Infected with M. marinum

A time series of the host transcriptional response of zebrafish larvae after infection with *M. marinum* was published [30], which allowed us to make a comparison between the response to infection with this surrogate model and the human pathogen. The *M. marinum* infection [30] was performed via manual intravenous injection of 28 hpf embryos and the time series included the late time points 2, 3, 4 and 5 dpi, which correspond with developmental time points 3, 4, 5 and 6 dpf, respectively. In addition, the *M. marinum*-infected larvae were incubated at 28.5 °C, whereas the Mtb-infected larvae were incubated at 34 °C. Since zebrafish embryonic development rate is temperature-dependent, we applied the formula H_T_ = h/(0.055T − 0.57), where H_T_ = hours of development at temperature T and h = hours of development to reach the stage at 28.5 °C [32], to determine which developmental stages were comparable at both temperatures. According to this calculation the 3, 4 and 5 dpi Mtb-infected larvae grown at 34 °C had developmental stages that were best comparable with, respectively, 3, 4 and 5 dpi (i.e., 4, 5, and 6 dpf) *M. marinum*-infected larvae grown at 28.5 °C. Statistical DESeq analysis of the RNAseq datasets of the *M. marinum*-infected larvae was repeated using our pipeline and resulted in 200, 803 and 1134 genes with significantly increased expression at 3 dpi, 4 dpi and 5 dpi, respectively (Figure 7). The number of genes with reduced expression was 38, 322 and 221, respectively (Figure 7). Pair-wise comparison of up-regulated genes in developmental stage-matched *M. tuberculosis*- and *M. marinum*-infected larvae showed an overlap of 10.6% (44 genes), 28.5% (81 genes) and 57.4% (187 genes) at 3 dpi, 4 dpi and 5 dpi, respectively (Figure 7). The identities of the 20 common genes in these three overlap sets are presented in Table 3. In addition, we found that 15 out of 23 genes that were always up-regulated in 3, 4, 5, 6 and 8 dpi; *M. tuberculosis*-infected larvae were also up-regulated at all three stages (3, 4, 5 dpi) of *M. marinum*-infected larvae, and an additional six genes were up-regulated in at least one of these stages (Table 2).

### 3.5. Comparison of Transcriptome Data of Zebrafish Mtb Infection with Human Macrophage Mtb Infection

We compared our transcriptome results of the core set of Mtb-induced host genes in *zebrafish* with previously published RNAseq data of infected human macrophages in our laboratories [31]. We compared the results with the overlap of genes responding to *M. tuberculosis* infection from human macrophage types MF1 and MF2 (Appendix A), of which the majority could be coupled to zebrafish orthologous gene codes. In these experiments 11 and 161 genes are commonly induced after 4 h or 24 h of infection, respectively. The results (Table 4) show that only a limited set of eight genes are up-regulated in the zebrafish and the 24 hr-infected macrophages. The human macrophages show a number of 148 and 438 commonly down-regulated genes after 4 or 24 h of infection, respectively. Of these genes, 28 genes are conservatively down-regulated in both data sets either in the 4 or 24 h macrophage infection data sets.

## 4. Discussion

Zebrafish is a natural host of various *Mycobacterium* species and a surrogate model organism for tuberculosis research. *M. marinum* is evolutionarily most closely related to *M. tuberculosis* and shares a majority of virulence genes. Although zebrafish is not a natural host of the human pathogen, we have previously demonstrated successful robotic infection of zebrafish embryos with *M. tuberculosis* and performed drug treatment of the infected larvae [20]. By comparing the responses of zebrafish larvae to infection of *M. marinum* and *M. tuberculosis*, we aim to further extend the use of this infection model as a stepping stone towards translational studies in other model organisms. In this study, we extended the previously performed study of *M. tuberculosis* infection in zebrafish larvae by extending the time scale of infection and including detailed transcriptome studies of the host responses. The results of the fluorescence microscopy and MBL assay indicate that zebrafish embryos are still infected with *M. tuberculosis* bacteria 9 days post-infection. This corroborated by the observation that after 8 days post-infection, there is still an active innate immune response at the transcriptional level. Although the quantification of the Mtb infection burdens in zebrafish using MBL at the stage up to 9 dpi holds little promise, it might be of use to study later stages of infection in follow up experiments directed at studying the function of the adaptive immune system in defense against tuberculosis in the zebrafish model.

A disadvantage of the *M. tuberculosis* zebrafish infection model is that it is dependent on maintaining the zebrafish larvae at a temperature of 34 °C. Our results show that incubation at 34 °C is very stressful to the zebrafish larvae and affects the expression of at least 15% of all annotated genes at all three time points (3, 4 and 5 dpi) that were examined. These changes include major effects on regulatory networks of the cytoskeleton and metabolism that are likely to change cellular host responses to infection. It is all the more surprising that, in this background of massive heat-stress-induced transcriptome changes, the innate immune host response against the two *Mycobacterium* species is very specific and seemingly unaffected by temperature. The results show that for functional analysis of responses to *M. tuberculosis* at later stages of infection the zebrafish larval model is extremely difficult. The results suggest that it is important to investigate the possibilities of other fish species that have a better tolerance to high temperatures as an alternative model for studying infection with *M. tuberculosis* at 34 °C.

We found that fifteen out of twenty-three genes that were always up-regulated in 3, 4, 5, 6 and 8 dpi *M. tuberculosis*-infected larvae were also up-regulated at all three stages (3, 4, 5 dpi) of *M. marinum*-infected larvae, and an additional six genes were up-regulated in at least one of these stages of infection (Table 2). When overlap data sets of only the three earliest time points of infection were compared, a very similar gene set was obtained (Table 3). However, at 8 dpi, there was still a strong increase in genes up or down-regulated in response to infection by *M. tuberculosis*. However, the induction of several inflammatory genes, such as *il1b*, was no longer detectable at this late stage. In future research, it will be interesting to compare these responses to response to infection with *M. marinum* at later time points than in published studies. The identified common response sets included several known immune response markers (Table 2). One of these markers, *dram1*, has been extensively studied in our laboratories for its function in autophagy [30]. This indication that *M. tuberculosis* bacteria in zebrafish larvae are also subject to defense by autophagy open future possibilities for comparison of autophagic responses to different classes of mycobacteria in one animal model organism. In addition to common responses, there are also specific responses of zebrafish larvae to *M. tuberculosis,* such as the gene *p2ry13* (purinergic receptor P2Y, G-protein coupled, 13). In addition, there are several genes in which the expression is greatly modulated by infection that have no known function, or are even completely not annotated (e.g., ENSDARG00000100518), but that could be important based on motives in their predicted encoded proteins. It will be of interest to further investigate whether orthologues of such genes also are induced in mammalian models of *M. tuberculosis* infection.

We compared our transcriptome results of the core set of Mtb-induced host genes in zebrafish with previously published RNAseq data of infected human macrophages from our laboratories [31]. Of the down-regulated genes that are common for both MF1- and MF2-infected macrophage signatures, the dual-specificity phosphatase 8 (*dusp8*) gene is standing out because of the observation that mouse Dusp proteins have been previously reported to have a role in promoting mouse macrophage autophagy to reduce mycobacteria survival via micro-RNA regulation [33]. Of the genes induced by Mtb in zebrafish larvae, we only found orthologous genes induced by Mtb infection in the MF2 cell population (Table 4). Several of the genes have been shown to be highly relevant for human macrophage infection. For instance, CD180 (also called RP105) interacts with Toll-like receptor TLR2 and facilitates the agonistic recognition of mature lipoproteins expressed by Mtb, such as the N-terminus of Mtb19 kDa lipoprotein [34]. CD180 is involved in engaging phosphatidylinositol 3-kinase p110δ in order to facilitate trafficking and secretion of cytokines in macrophages during Mtb infection [35]. Another induced gene that has been shown to be important for macrophage infection is the gene encoding the phospholipid-binding protein annexin. For instance, the study of Gan et al. [36] shows that Mtb blocks crosslinking of annexin-1 and apoptotic envelope formation on infected macrophages to maintain virulence. Since zebrafish genes are often duplicated compared to mammalian counterparts, this will, in several cases, require further study of orthology relationships to confirm the translational value of our findings for identifying new tuberculosis diagnostic markers [37,38]. In conclusion, our results confirm the systemic relevance of several genes involved in macrophage defense in mammalian systems. Further investigation of the other identified marker genes is therefore of general interest. Considering the high throughput capacity of our robotic injection system, we envisage the application for discovering additional tuberculosis diagnosis genes, but also the screening of new anti-tuberculosis drugs via host-directed targeting (HDT). Our transcriptomic data set provides a highly valuable reference for future studies using the *M. marinum* model that, at this stage, is far superior for functional studies of mycobacterial infection than the *M. tuberculosis* infection model due to the technical limitations of temperature and high biosafety protocols. Furthermore, we expect that the *M. tuberculosis* model can be used to confirm outcomes of such functional studies using *M. marinum* at the transcriptome level.

## 5. Conclusions

This study underscores the potential of zebrafish larvae as a valuable model for investigating transcriptomic effects of *M. tuberculosis* infection, a significant cause of human tuberculosis. Despite zebrafish larvae not being natural hosts for *M. tuberculosis*, successful infection and propagation were achieved for up to 9 days post-injection, facilitated by an efficient robotic system. Molecular bacterial load assays and transcriptome analysis confirmed the persistence of viable *M. tuberculosis* in the infected zebrafish larvae and unveiled a specific transcriptional immune response, akin to the response induced by *M. marinum.* Therefore, this research supports the endorsement of the *M. marinum* infection model as a robust surrogate for tuberculosis studies. By comparing key immune response genes induced by *M. tuberculosis* in zebrafish larvae with those triggered by human macrophages, the study revealed shared and unique gene expression patterns. The findings contribute valuable insights to the development of preclinical tools for tuberculosis research using the *M. marinum* model, emphasizing the potential of the zebrafish model for identifying diagnostic markers of *M. tuberculosis* infections in mammals and screening potential drugs.

## Figures and Tables

**Figure 1 biology-13-00688-f001:**
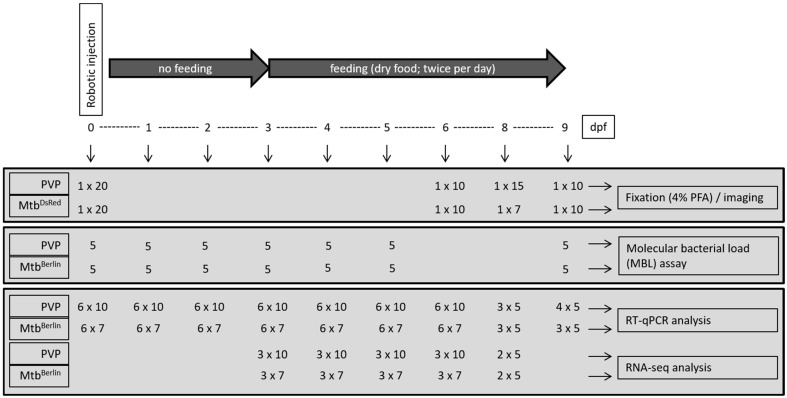
Experimental design. Used was the strain *M. tuberculosis* H37Rv (Mtb Berlin) without plasmid obtained from the Kaufman lab, MPI, Berlin, Germany. In the case of the fluorescent imaging and survival curves an additional strain containing a DsRed expressing plasmid (Mtb^DsRed^) was used. Numbers indicate the number of replicates followed (after symbol x) by the number of larvae per replicate. Vertical arrows indicate the time points of sampling.

**Figure 2 biology-13-00688-f002:**
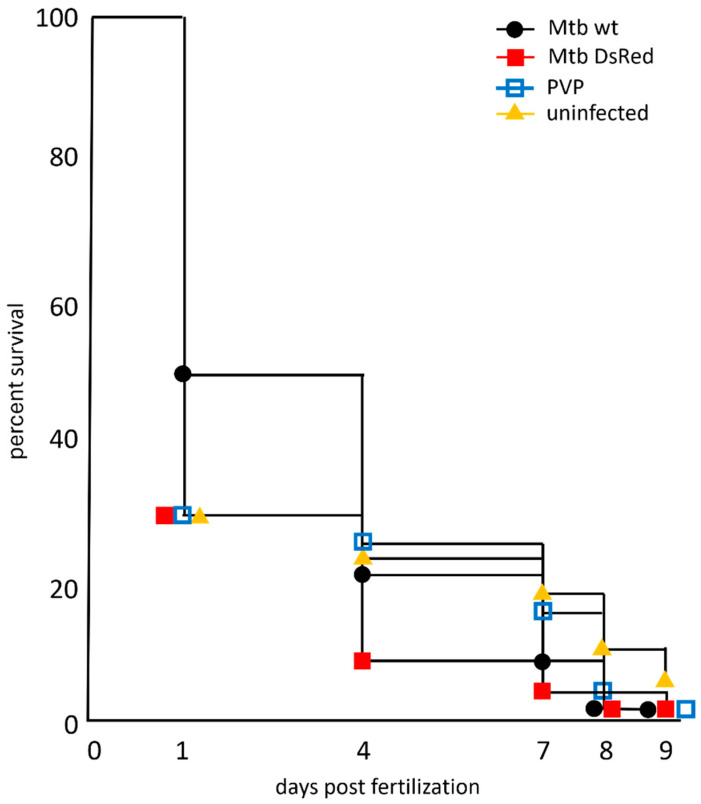
Survival of zebrafish larvae at 34 °C upon *M. tuberculosis* infection. Zebrafish embryos between 16 and 128-cell stage were robotically infected with 500 CFU *M. tuberculosis* wild-type (
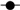
) or *M. tuberculosis*-DsRed (
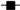
)H37Rv strains and raised at 34 °C along with control groups of PVP-injected (

) and uninjected (
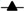
) embryos. The last viable larvae were collected at 9 dpf. Percent survival for each group is shown on the survival curve. The number of injected embryos was 1000, 1200, 2000 and 400 for the *M. tuberculosis* wild-type strain, *M. tuberculosis*-H37Rv DsRed strain, PVP-injected and uninjected embryos, respectively. Raw data is presented in Appendix A.

**Figure 3 biology-13-00688-f003:**
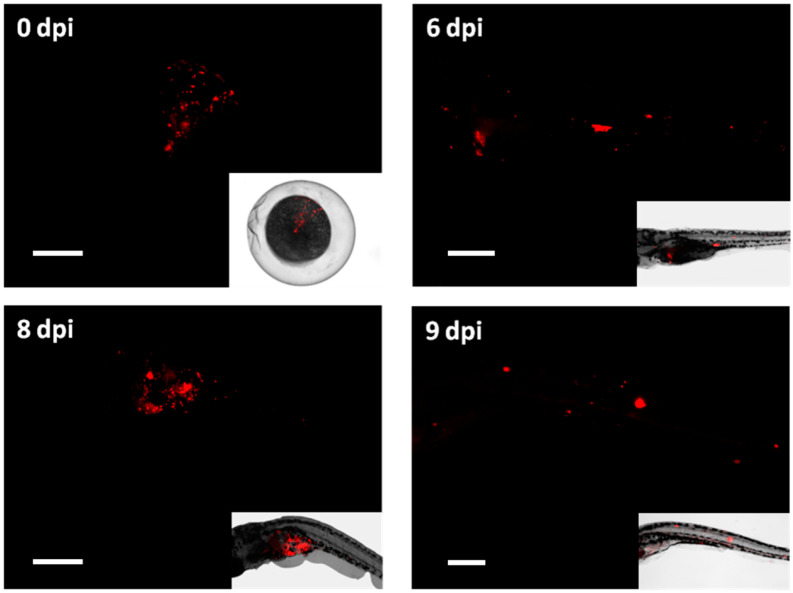
Fluorescent microscopy imaging of *Mycobacterium tuberculosis* infection in zebrafish larvae. Images taken with a fluorescent stereomicroscope of 0, 6, 8 and 9 dpi *M. tuberculosis*-DsRed infected larvae are shown together with the bright field image of the same embedded larvae. Image of 0 dpi embryo represents a successful injection of fluorescent bacteria into the yolk. At 6, 8 and 9 dpi bacteria showed dissemination throughout the body of the larvae and the formation of bacterial aggregates. Scale bar: 200 micrometer.

**Figure 4 biology-13-00688-f004:**
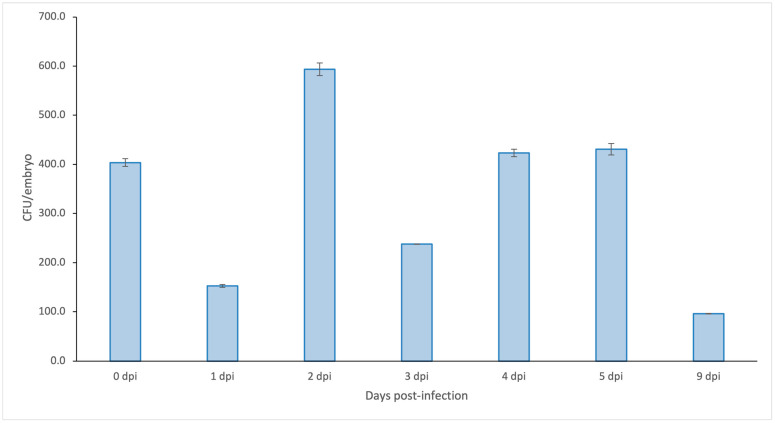
*M. tuberculosis* CFU counts per embryo based on molecular bacterial load (MBL) assay. Zebrafish embryos were infected with *M. tuberculosis* H37Rv at 0 dpf. Total RNA was purified at 0, 1, 2, 3, 4, 5 and 9 dpi and used as template in the MBL assay Mtb-specific 16S rRNA in a RT-qPCR. The figure shows the amount of CFU per embryo. Raw data are provided in Appendix A.

**Figure 5 biology-13-00688-f005:**
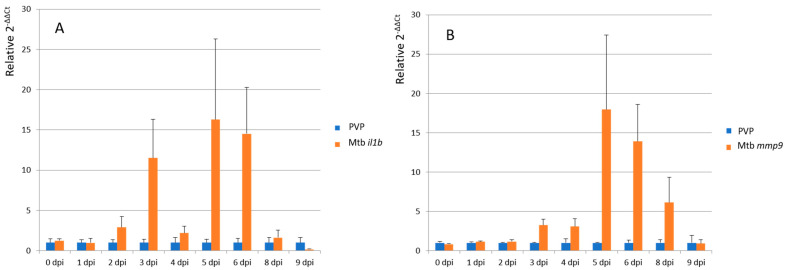
Differential gene expression of *il1b* (**A**) and *mmp9* (**B**). Differential gene expression measured by qPCR of cDNA for RNA isolated from zebrafish larvae infected with *M. tuberculosis* (Mtb) at different times after injection (dpi). Strain H37Rv without a plasmid was used for these experiments. A comparison is made with the mock-injection results with the carrier PVP. The raw data and statistical analysis is presented in Appendix A.

**Figure 6 biology-13-00688-f006:**
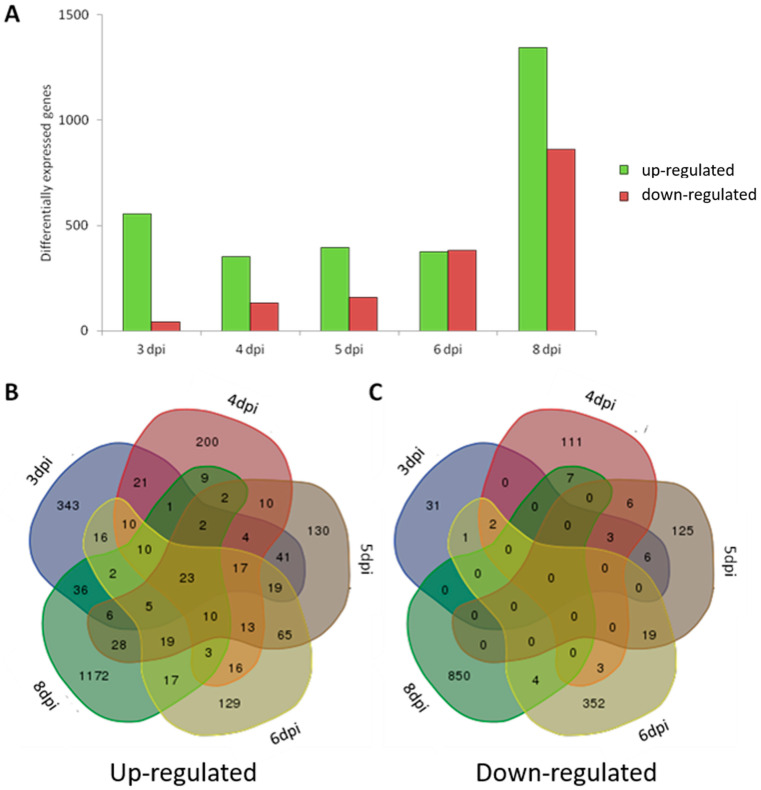
Differentially expressed genes in the time-course infection experiment with *Mycobacterium tuberculosis*. (**A**) The total number of significantly differentially expressed genes at 3, 4, 5, 6, and 8 dpi larvae infected with *Mycobacterium tuberculosis* compared to the PVP-injected control group. Strain H37Rv without a plasmid was used for these experiments. Significance was set to a fold change larger than 2 for up-regulated genes (green bars) or smaller than −2 for down-regulated genes (red bars), with a *p* value smaller than 0.05. (**B**,**C**) Venn diagrams show the overlap of the significantly up-regulated (**B**) or down-regulated (**C**) genes between the five time points of mycobacterium infection.

**Figure 7 biology-13-00688-f007:**
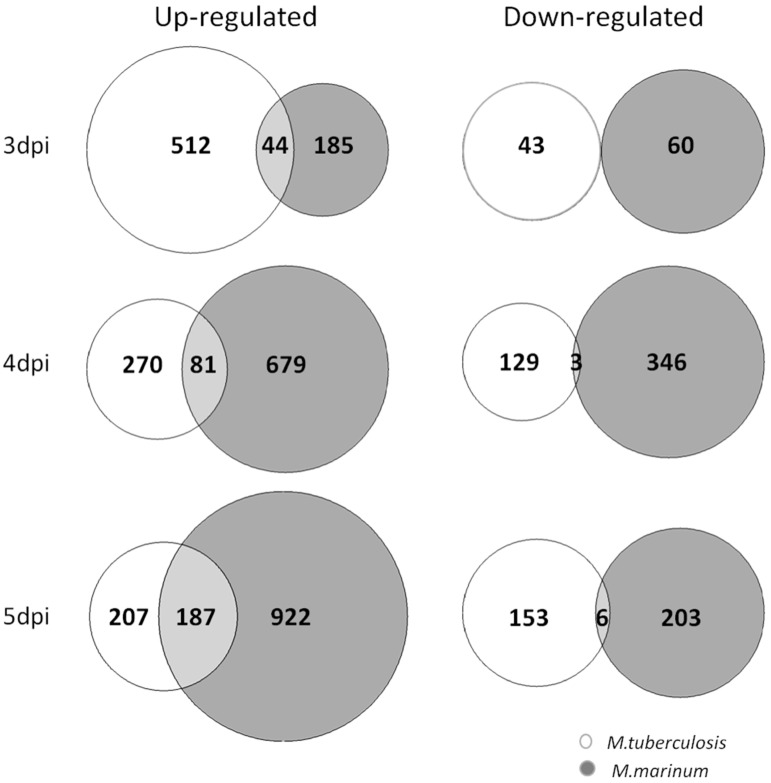
Comparison of differentially expressed host genes in *M. tuberculosis* and *M. marinum* infection experiments. The expression profiles are compared after aligning the developmental stages of larvae grown at an increased temperature (34 °C) in the *M. tuberculosis* (Mtb) infection experiment with the ones developed under normal condition (28 °C) in the *M. marinum* (Mm) infection. Area-proportional Venn diagrams visualize the number of the significantly up- and down-regulated genes in the stage-matched larvae of *M. tuberculosis* (white) and *M. marinum* (gray) infection. The 3 dpi Venn diagrams represent 3 dpi (3 dpf) Mtb vs. 3 dpi (4 dpf) Mm data set, the 4 dpi Venn diagrams represent 4 dpi (4 dpf) Mtb vs. 4 dpi (5 dpf) Mm data set, and the 5 dpi Venn diagrams show 5 dpi (5 dpf) Mtb vs. 5 dpi (6 dpf) Mm data set.

**Table 1 biology-13-00688-t001:** Comparison of *M. tuberculosis* and *M. marinum* [5].

Mycobacterium Species	*M. tuberculosis*	*M. marinum*
natural hosts	humans, cattle	fish, amphibians, zoonosis
infection route	lung	skin
optimum temperature	38 °C	32 °C
minimum doubling time	18 h	4 h
biosafety level	3	2
genome size	4.41 Mb	6.64 Mb
gene number	4008	5503

**Table 2 biology-13-00688-t002:** Core set of 23 genes that is always up-regulated in *M. tuberculosis*-infected larvae at 3, 4, 5, 6 and 8 dpi and overlap with up-regulated genes in Mm-infected larvae at 3, 4, and 5 dpi (4, 5 and 6 dpf). Indicated is whether genes are upregulated genes (√) or not signficant changed (-).

Gene Code	Gene Name	Gene Description	Mm 3 dpi	Mm 4 dpi	Mm 5 dpi
ENSDARG00000101479	*BX908782.3*	three-finger protein 5 (LOC100003647)	√	√	√
ENSDARG00000093712	*CABZ01021530.1*	Interferon-inducible protein Gig1-like (teleost-specific)	√	√	√
ENSDARG00000086654	*cbln11*	cerebellin 11	√	√	√
ENSDARG00000026417	*ccr12b.2*	chemokine (C-C motif) receptor 12b2	√	√	√
ENSDARG00000051912	*hpx*	hemopexin	√	√	√
ENSDARG00000069844	*irg1*	immunoresponsive 1 homolog (aconitate decarboxylase)	√	√	√
ENSDARG00000003523	*itln3*	intelectin 3 (bacterial arabinogalactan receptor)	√	√	√
ENSDARG00000090889	*lect2*	leukocyte cell-derived chemotaxin 2	√	√	√
ENSDARG00000012395	*mmp13a*	matrix metallopeptidase 13a	√	√	√
ENSDARG00000042816	*mmp9*	matrix metallopeptidase 9	√	√	√
ENSDARG00000045999	*saa*	serum amyloid A	√	√	√
ENSDARG00000086337	*si:dkey-102g19.3*	urokinase plasminogen activator surface receptor-like	√	√	√
ENSDARG00000095798	*si:dkey-119g10.3*	non-protein coding	√	√	√
ENSDARG00000090352	*si:dkey-97i18.5*	non-protein coding	√	√	√
ENSDARG00000007769	*sult5a1*	sulfotransferase family 5A, member 1	√	√	√
ENSDARG00000045561	*dram1*	DNA-damage regulated autophagy modulator 1	-	√	√
ENSDARG00000026049	*mxf*	myxovirus (influenza virus) resistance F	-	√	√
ENSDARG00000095909	si:ch211-253b18.3	non-protein coding	-	√	√
ENSDARG00000086947	si:ch211-147m6.1	olfactomedin-4-like	√	-	-
ENSDARG00000104399	*CABZ01064972.2*	Uncharacterized protein	-	-	√
ENSDARG00000033587	*CABZ01088134.1*	trans-2-enoyl-CoA reductase, mitochondrial-like	-	-	√
ENSDARG00000089362	*grn1*	granulin 1	-	-	√
ENSDARG00000069944	*p2ry13*	purinergic receptor P2Y, G-protein coupled, 13	-	-	-

**Table 3 biology-13-00688-t003:** Commonly regulated genes at 3, 4 and 5 dpi time points of both *M. tuberculosis* and *M. marinum* infection. Due to lack of commonly down-regulated genes only an up-regulated list is presented.

	Gene Name	Description
Up-Regulated		
ENSDARG00000067672	*card9*	caspase recruitment domain family, member 9
ENSDARG00000090873	*ccl34a.4*	chemokine (C-C motif) ligand 34a, duplicate 4
ENSDARG00000026417	*ccr12b.2*	chemokine (C-C motif) receptor 12b, tandem duplicate 2
ENSDARG00000055278	*cfb*	complement factor B
ENSDARG00000051912	*hpx* (1 of many)	hemopexin
ENSDARG00000053131	*irak3*	interleukin-1 receptor-associated kinase 3
ENSDARG00000069844	*irg1*	immunoresponsive 1 homolog (mouse)
ENSDARG00000003523	*itln3*	intelectin 3
ENSDARG00000090889	*lect2* (1 of many)	leukocyte cell derived chemotaxin 2
ENSDARG00000012395	*mmp13a*	matrix metallopeptidase 13a
ENSDARG00000042816	*mmp9*	matrix metallopeptidase 9
ENSDARG00000033735	*ncf1*	neutrophil cytosolic factor 1
ENSDARG00000045999	*saa*	serum amyloid A
ENSDARG00000053836	si:ch211-284o19.8	si:ch211-284o19.8
ENSDARG00000086337	si:dkey-102g19.3	si:dkey-102g19.3
ENSDARG00000095798	si:dkey-119g10.3	si:dkey-119g10.3
ENSDARG00000097909	si:dkey-195m11.11	si:dkey-195m11.11
ENSDARG00000090352	si:dkey-97i18.5	si:dkey-97i18.5
ENSDARG00000055252	*snap23.2*	synaptosomal-associated protein 23.2
ENSDARG00000007769	*sult5a1*	sulfotransferase family 5A, member 1

**Table 4 biology-13-00688-t004:** Comparison of the host transcriptome data from this study and previously published data from infection of human macrophages with *M. tuberculosis*. The data from the genes that is up-regulated (2379 genes) or down-regulated (1512 genes) in *M. tuberculosis* (Mtb)-infected larvae at 3, 4, 5, 6 and 8 dpi is compared with the transcriptome data set from human M1 and M2 macrophages infected with Mtb published by Vrieling et al. [31]. *: An asterisk indicates genes that are also down regulated after *M. marinum* infection at 5 dpi.

Up-Regulated Genes					
Zebrafish (Mtb3,4,5,6,8-2379 Genes)	Human Macrophages (MF1 and 2)		Ensembl Gene ID	Gene Name	Description
	4 hr (11 Genes)	24 hr (161 Genes)			
x (Mtb3)		x	ENSDARG00000002917	*gls2b*	glutaminase 2b (liver, mitochondrial)
x (Mtb3)		x	ENSDARG00000093750	*cd180*	CD180 molecule
x (Mtb5)		x	ENSDARG00000036106	*rgs18*	regulator of G-protein signaling 18
x (Mtb5)		x	ENSDARG00000101050	*tmc8*	transmembrane channel-like 8
x (Mtb8)		x	ENSDARG00000009978	*icn*	ictacalcin
x (Mtb8)		x	ENSDARG00000013335	*anxa6*	annexin A6
x (Mtb8)		x	ENSDARG00000087554	*cdk1*	cyclin-dependent kinase 1
x (Mtb8)		x	ENSDARG00000098946	*pald1a*	phosphatase domain containing, paladin 1a
**Down-Regulated Genes**					
**Zebrafish (Mtb3,4,5,6,8-1520 Genes)**	**Human Macrophages (MF1 and 2)**		**Ensembl Gene ID**	**Gene Name**	**Description**
	**4 hr (149 Genes)**	**24 hr (438 Genes)**			
x (Mtb6)	x		ENSDARG00000062947	*amn*	amnion associated transmembrane protein
x (Mtb8)	x		ENSDARG00000030722	*xirp1*	xin actin binding repeat containing 1
x (Mtb8)	x		ENSDARG00000037196	*arid5B*	AT-rich interaction domain 5B
x (Mtb8)	x		ENSDARG00000056929	*kdm6bb*	lysine (K)-specific demethylase 6B, b
x (Mtb8)	x		ENSDARG00000070538	*hey1*	hes-related family bHLH transcription factor with YRPW motif 1
x (Mtb8)	x		ENSDARG00000105261	*nfkb1*	nuclear factor of kappa light polypeptide gene enhancer in B-cells 1
x (Mtb5)		x	ENSDARG00000102729	*samd9*	sterile alpha motif domain containing 9
x (Mtb5)		x	ENSDARG00000016733	*psat1*	phosphoserine aminotransferase 1
x (Mtb8)		x	ENSDARG00000005526	*igfn1.1*	immunoglobulin-like and fibronectin type III domain containing 1, tandem duplicate 1
x (Mtb8)		x	ENSDARG00000011257	*enpp2*	ectonucleotide pyrophosphatase/phosphodiesterase 2
x (Mtb8)		x	ENSDARG00000018773	*hivep2* (1 of many)	human immunodeficiency virus type I enhancer binding protein 2
x (Mtb8)		x	ENSDARG00000020581	*otofb*	otoferlin b
x (Mtb8)		x	ENSDARG00000040523	*smpd2a*	sphingomyelin phosphodiesterase 2a, neutral membrane (neutral sphingomyelinase)
x (Mtb8)		x	ENSDARG00000042055 *	*fam129aa*	family with sequence similarity 129, member Aa
x (Mtb8)		x	ENSDARG00000045802	*hapln3*	hyaluronan and proteoglycan link protein 3
x (Mtb8)		x	ENSDARG00000057317	*nexn*	nexilin (F actin binding protein)
x (Mtb8)		x	ENSDARG00000059090	*sstr2* (1 of many)	somatostatin receptor 2
x (Mtb8)		x	ENSDARG00000061070	*chst3a*	carbohydrate (chondroitin 6) sulfotransferase 3a
x (Mtb8)		x	ENSDARG00000063475	*abcg1*	ATP-binding cassette, sub-family G (WHITE), member 1
x (Mtb8)		x	ENSDARG00000075263	*ankrd1a*	ankyrin repeat domain 1a (cardiac muscle)
x (Mtb8)		x	ENSDARG00000088137	*adrg2* (1 of many)	adhesion G protein-coupled receptor G2
x (Mtb8)		x	ENSDARG00000088937	*adrg2* (1 of many)	adhesion G protein-coupled receptor G2
x (Mtb8)		x	ENSDARG00000100518	Uncharacterized	
x (Mtb8)	x	x	ENSDARG00000007080 *	*rhcgl1*	Rh family, C glycoprotein, like 1
x (Mtb8)	x	x	ENSDARG00000013168	*jag1b*	jagged 1b
x (Mtb8)	x	x	ENSDARG00000039232	*dusp8* (1 of many)	dual specificity phosphatase 8
x (Mtb8)	x	x	ENSDARG00000089871	*si:ch211-203c7.2*	si:ch211-203c7.2

## Data Availability

The RNAseq data are submitted in the NCBI GEO database as project GSE252417, Acc. Nrs. GSM8001074-GSM8001105. The raw data of the molecular bacterial load (MBL) and quantitative RT-PCR (Q-PCR) are available in the Appendix A.

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
