# Peer review of "The Human Pathogen Mycobacterium tuberculosis and the Fish Pathogen Mycobacterium marinum Trigger a Core Set of Late Innate Immune Response Genes in Zebrafish Larvae"

_biology, 2024, doi:10.3390/biology13090688_

Round 1
Reviewer 1 Report (New Reviewer)
Comments and Suggestions for Authors
The current study investigates the use of zebrafish as a model organism for tuberculosis infection research, focusing on the infection of zebrafish larvae with Mycobacterium tuberculosis. Although zebrafish are not natural hosts for M. tuberculosis, previous work from the same authors has shown successful infection and drug treatment of zebrafish embryos with this bacterium. Here, the authors focus on the duration that M. tuberculosis can survive in zebrafish larvae and use RNA sequencing to analyze the host transcriptional response over time. Their findings suggest that M. tuberculosis can persist in zebrafish larvae for at least a week, providing a valuable model for tuberculosis research. Overall the study presented here has potential to contribute to developing new preclinical models for TB research.
However, there are some major issues that need to be improved, before I can recommend this article for publication:
-
It seems to me that the observed mortality is due to temperature and not to the bacterial infection, since there are really no differences in mortality between infected and non-infected groups. Moreover, the only slight difference is on day 1 and is the infected group the one with higher survival rate. Thus, the authors should include a temperature control (embryos raised at 28C) for comparison reasons. This might shed light on the mortality reasons and demonstrate that temperature is an important limitation for this model, reinforcing their claim of M. marinum value as a surrogate model for TB-ZF interactions.
-
Given the variability of results of Mtb presence on the larvae between methods used for quantification, the high mortality found in control groups, and the fact that MBL quantification (which seems to be the method given better results) has only being performed once, this study cannot be published as it is. The authors should repeat at least the MBL assay twice more, to ensure their results. Also, the Mtb strains with and without plasmid seem to behave differently (see mortality results), so this might be influencing the variable results obtained for the different methods used to assess the time Mtb persists in the larvae (since different strains have been used for different methods). The authors should re-do the experiments all with the fluorescent strains for comparison reasons.
-
Overall, it seems that the major take away from this study is that i) zebrafish infection with M. tuberculosis is not really a good model due to incubation temperature restrictions; ii) the results obtained here and in previous studies suggest that M. marinum infection of zebrafish can be a good surrogate preclinical model for TB research. However, the way the article is written suggests that the authors are encouraging to use Mtb as the infecting organism, even if the infection model has a few major issues to trust the infection. Thus, the authors should rewrite the article focussing on highlight the strengths of M. marinum-ZF interaction as a TB infection model.
-
I do not understand the numbers in the figure (i.e., 1 x 20). Do they refer to the number of groups x number of larvae per group? If yes please specify it in the caption, if not please explain further. Also, in Figure 1, please include the MLB assay in the experimental design.
-
Lines 316-318: Given the difference in fold increase between 3 and 4 dpi for il1b expression, it doesn't seem logical to combine these samples for statistical analysis, the authors should consider including more biological replicates instead.
-
Figure 7: A Venn diagram comparing the transcriptomes at different temperatures (section 3.3) would also help to interpret this results, since the majority of the differences, as stated by the authors, seem to be temperature-dependent and not pathogen-dependent. This will reinforce the idea that the core DE genes are the ones relevant for the infection and that M. marinum is a good model.
-
Line 176: Figure 8 is missing in the paper.
Specific minor corrections:
-
Some sentences are off, should be revised and rewritten: i.e., lines 32-34; line 281;
-
Keywords are too vague, they should be more specific.
-
In line 140, 1 nl bacterial suspension, means nm? It is unclear, please revise.
-
Line 238: Mtb. You have previously stated this in line 107, please remove it from here and stick to the abbreviation throughout the manuscript. It is a bit confusing that the abbreviation is sometimes used and sometimes not.
-
Figure 1 is nor referenced in the text, please do so.
-
Line 297: stereomicroscope of
-
Figure 4: Specify: il1b (A) and mmp9 (B) in the caption so it’s easier to identify the data in the figure.
-
Lines 142 - 143: Check and re-write
Author Response
See attachment

Reviewer 2 Report (New Reviewer)
Comments and Suggestions for Authors
Author Response
Please see the attachment

Reviewer 3 Report (New Reviewer)
Comments and Suggestions for Authors
The authors of the submitted paper present an unusual model of experimental tuberculosis. The article describes the interaction of zebrafish larvae with Mycobacterium tuberculosis, an atypical parasite for this organism. Methods used are appropriate to study objectives. Under atypical conditions for both host and parasite, the study examines the development of an immune response against M. tuberculosis. The authors compare the expression profiles of M. tuberculosis infected zebrafish with those of M. marinum infected zebrafish. Common patterns of gene expression are identified by comparing the response of the zebrafish model to M. tuberculosis with that of human macrophages. Based on the analysis of the results, it is concluded that zebrafish can be further used as a host model for the identification of diagnostic markers of M. tuberculosis infection and for the screening of potential drugs.
This is probably too bold a statement. The zebrafish model of mycobacterial infection is certainly very interesting. However, it is too far removed from human TB, which is important for identifying diagnostic markers and for HDT. This model has more disadvantages (atypical, complicated, expensive) than advantages. The abstract and conclusion of the paper need to be adjusted.
Author Response
The authors of the submitted paper present an unusual model of experimental tuberculosis. The article describes the interaction of zebrafish larvae with Mycobacterium tuberculosis, an atypical parasite for this organism. Methods used are appropriate to study objectives. Under atypical conditions for both host and parasite, the study examines the development of an immune response against M. tuberculosis. The authors compare the expression profiles of M. tuberculosis infected zebrafish with those of M. marinum infected zebrafish. Common patterns of gene expression are identified by comparing the response of the zebrafish model to M. tuberculosis with that of human macrophages. Based on the analysis of the results, it is concluded that zebrafish can be further used as a host model for the identification of diagnostic markers of M. tuberculosis infection and for the screening of potential drugs.
This is probably too bold a statement. The zebrafish model of mycobacterial infection is certainly very interesting. However, it is too far removed from human TB, which is important for identifying diagnostic markers and for HDT. This model has more disadvantages (atypical, complicated, expensive) than advantages. The abstract and conclusion of the paper need to be adjusted.
Response:
We do agree that for screening the M.tuberculosis zebrafish infection model is still not up to a high through put through put level. We therefore have adjusted several sentences to make this clear. However, every other mammalian animal model also suffers from a lack of through put under high safety conditions. Therefore for in depth screening we recommend here the zebrafish M.marinum infection model at high throughput and other models, such as the one described in this paper (and subsequently mammalian models or cell culture models) for further translation of these findings. Rodent models also have a temperature difference compared to human infection so therefore comparative models with different temperatures remain relevant. Primate studies are certainly the best animal models for studying TB but these are highly restricted in The Netherlands.
Reviewer 4 Report (New Reviewer)
Comments and Suggestions for Authors
This paper from Dirks and colleagues reports on the use of zebra-fish larvae as a model host for studying aspects of M. tuberculosis infection. They compare their results to published results with M. marinum, a fish pathogen that has many of the same virulence genes as M. tuberculosis. This is somewhat amusing, because the rationale for the M. marinum model was as a surrogate for studying M. tuberculosis. Of great interest, they suggest this could be a way of testing new antimicrobial drugs in vivo.
They use Ds-red labelled M. tuberculosis to visualize the spread of bacteria after injection into the yolk sac, and unlabeled H37Rv to study the kinetics of growth of bacteria after injection over an 8 day period. They also collect the surviving larvae to extract total mRNA to detect changes in gene expression over time. All of these methods are clearly described.
Comments.
1. Overall, the English is clear, but there are examples of odd word choices, e.g.: on page 1 showcasing line 29; endorses on line 35; host model rather than model host on line 39.
2. Since they decided to grow the fish at 34C they introduce the complication of heat stress. Although they are aware of this and know what host genes were upregulated by temperature, they do not now or comment on how those changes might influence the response to infection.
3. Table 1 compares M. marinum and M. tuberculosis but makes no mention of ESAT genes or secretion systems in marinum. I don’t think this table adds anything to what the text says.
4. Figure 1 is a graphic of the timeline of the experiment but there is no explanation of what the vertical arrows mean, nor what all the numbers refer to e.g. 1x10.
5. The mortality rate from the initial injection is about 70% by day 1 (figure 2). The number of embryos in each group is not stated. Since they claim robotic injection of the yolk sac is better than hand injection, what would the mortality rate be for hand injections?
6. Given the small percentage of surviving larvae by day 8 and 9, I am skeptical about their inclusion in figure 4 and 5. What was the N on those days?
7. Figure 3 is very nice but little to interpret the anatomy of distribution on days 6,8, and 9. Can they elaborate? In the legend they refer to clumps as granuloma-like, but granulomas are a host response to a foreign material, and they are not visualizing host cells, so this could just as easily be the hydrophobic bacteria clumping together, or several bacteria inside one host cell.
8. The literature about zebra fish infections frequently refers to the host response, suggesting it is analogous to human responses, but this is a model of infection of immature macrophages, and most of the measured responses they highlight are macrophage response genes. There are no lymphocytes involved in modulating macrophages in these embryos.
9. They suggest that this may be a good model to test the efficacy of new drug in vivo, but what about pharmacokinetics in the embryos compares to humans?
10. Table 2 needs to be put back together (it is divided on 2 pages).
11. References 13-15 do not support the statement that M. marinum causes fatal infections in PLWAIDS.
Comments on the Quality of English Language/
Round 2
Reviewer 1 Report (New Reviewer)
Comments and Suggestions for Authors
The current study investigates the use of zebrafish as a model organism for tuberculosis infection research, focusing on the infection of zebrafish larvae with Mycobacterium tuberculosis. Although zebrafish are not natural hosts for M. tuberculosis, previous work from the same authors has shown successful infection and drug treatment of zebrafish embryos with this bacterium. Here, the authors focus on the duration that M. tuberculosis can survive in zebrafish larvae and use RNA sequencing to analyze the host transcriptional response over time. Their findings suggest that M. tuberculosis can persist in zebrafish larvae for at least a week, providing a valuable model for tuberculosis research. Overall the study presented here contributes to developing new preclinical models for TB research.
In the revised version of this manuscript the authors have made imrpovements according to the suggestions (i.e. Figure 1 improvement and all the small corrections suggested), but there are still some unsolved major issues that prevent me to accept this paper for publication:
- It seems to me that the observed mortality is due to temperature and not to the bacterial infection, since there are really no differences in mortality between infected and non-infected groups. Moreover, the only slight difference is on day 1 and is the infected group the one with higher survival rate. Thus, the authors should include a temperature control (embryos raised at 28C) for comparison reasons. This might shed light on the mortality reasons and demonstrate that temperature is an important limitation for this model, reinforcing their claim of M. marinum value as a surrogate model for TB-ZF interactions.
Response:
We completely agree with this statement. This is also what we tried to express in the manuscript. We have made the text somewhat more explicit to state this more clearly. We have now included a control experiment with larvae from the same fish line at 28 degrees. This experiment was part of a larger unpublished experiment on the effect of antibiotics on treatment of infection with M. marinum performed by this team. As expected there is little effect of infection with M. marinum at 28 degrees provided that the larvae are treated with antibiotics. This shows that our robotic injection method and set up does not influence survival at 28 °C up to 7 dpi. Therefore it is clear that after 7 dpi the number of surviving larvae at the growth temperature of 34°C becomes limiting for analysis of responses to infection with M. tuberculosis, in contrast to studies with M.marinum at 28 °C. We have added an extra panel in Fig S1 and explained the additional experiment in the results and legend.
How do the authors explain the high mortality observed in their DMSO control during the experiment at 28ºC? It seems that this mortality is actually very simmilar to the PVP control at 7 dpi in the experiment at 34ºC, which male me suspect that if the experiment is extended 2 more days moratity will be simmilar to the final mortality at 9 dpi in the original experiment. Thus, if the mortality in controls groups is so high, how can drive conclusions on the role of infection with any of the strains in moratlity, etc.?
- Given the variability of results of Mtb presence on the larvae between methods used for quantification, the high mortality found in control groups, and the fact that MBL quantification (which seems to be the method given better results) has only being performed once, this study cannot be published as it is. The authors should repeat at least the MBL assay twice more, to ensure their results. Also, the Mtb strains with and without plasmid seem to behave differently (see mortality results), so this might be influencing the variable results obtained for the different methods used to assess the time Mtb persists in the larvae (since different strains have been used for different methods). The authors should re-do the experiments all with the fluorescent strains for comparison reasons.
Response:
It has to be stated that the ethical guidelines for animal testing in the Netherlands do not permit repetitive experiments just for the sake of repetition and without answering a scientific open question for a study proposal. This appears to be the case with this request of the reviewer for the following reasons: We concluded with this experiment that Mtb survives in zebrafish larvae and that conclusion is qualitatively validated. However, a quantitation of microbial survival is not a research question in this paper. So there can not be a license obtained to repeat these experiments. We provide all data not to answer a research question but for reproducibility of the experiments for other research questions such as the transcriptome research described in this paper. Anyway, to repeat the experiment can only be expected to show moderate statistics on the quantity of how much Mtb bacteria survive in zebrafish larvae. The number of bacteria is clearly very low and this is confirmed with the fluorescence microscopic analysis. This would mean that with any method further repetition would remain statically unsatisfying. This is actually a major reason why functional studies for tuberculosis in zebrafish should be, performed with M. marinum and not with M. tuberculosis. We have added a sentence to this effect in the conclusions and also have given more attention to this point in the discussion. As mentioned in our reply to point 1 of this reviewer, we do agree that there is no reason to suspect an effect of a plasmid with fluorescence since this has never been reported to be the case before (see for instance reference 20). Therefore, to repeat the experiment with a fluorescent strain answers no scientific question and in fact would add uncertainty to the study since the wild type strain is the standard for all research. Instead, we would like to stress that the transcriptome experiments that have been statistically well analysed show a transcriptional response to the Mtb bacteria enforcing our statement that the bacteria are functionally present in the fish at all stages of analyses. For detailed analysis of survival of Mtb in a fish model we recommend to start an investigation into other fish species that survive better at higher growth temperatures. We have added extra sentence in the discussion.
I understand the authors concern and the ethic regulations for animals experiments and I consider their response valid. Nevertheless, in my oponion it is important that they show why this experiments has not been repeated to anwer the quiestions. Thus I suggest that a justification for it is also included in the main text. This also applies for the qPCR results.
- Overall, it seems that the major take away from this study is that i) zebrafish infection with M. tuberculosis is not really a good model due to incubation temperature restrictions; ii) the results obtained here and in previous studies suggest that M. marinum infection of zebrafish can be a good surrogate preclinical model for TB research. However, the way the article is written suggests that the authors are encouraging to use Mtb as the infecting organism, even if the infection model has a few major issues to trust the infection. Thus, the authors should rewrite the article focussing on highlight the strengths of M. marinum-ZF interaction as a TB infection model.
Response:
We agree with the reviewer. We thank the reviewer for making us realize that our text could be interpreted in a way that was not our intention. We have now extended and rewritten parts of the discussion and conclusions and thereby follow the suggestions made by the reviewer.
I appreciate the answer from the authors, but I don´t think the paper reads better after the few lines they have added. I still have the same impression of misconfussion of the scientific question they want to answer, the experimental design proposed and the interpretations and discussion of the results not being in the same line.
Overall my suggestions is that the authors re-consider the way they present the results they have given the impossibility of repeating some of them to re-inforce their points, and this way solve the flaws in methodology. Simmilarly, the flow of the paper need to be re-consider to avoid misconceptions of their main findings.
Author Response
How do the authors explain the high mortality observed in their DMSO control during the experiment at 28ºC? It seems that this mortality is actually very simmilar to the PVP control at 7 dpi in the experiment at 34ºC, which male me suspect that if the experiment is extended 2 more days moratity will be simmilar to the final mortality at 9 dpi in the original experiment. Thus, if the mortality in controls groups is so high, how can drive conclusions on the role of infection with any of the strains in moratlity, etc.?
We thank the reviewer for this comment since it shows that it is important to give more information on this experiment in the legend. We have now added the follow description in the legend that makes the meaning of this control experiment more clear.
At the start of the experiment the larvae were supplied with 200 uM rifampicin and 2 mM Isoniazid in DMSO. As a control the solvent DMSO was added to the medium at a concentration of 0.4%. The mortality in the control due to the infection by M.marinum is conform earlier published results from our group [30]. The rescue by 200 uM rifampicin and 2 mM Isoniazid shows that there is no notable mortality due to the injection procedure or quality of the larvae. Therefore the effects we observe at 34 degrees is attributed to the temperature.
I understand the authors concern and the ethic regulations for animals experiments and I consider their response valid. Nevertheless, in my oponion it is important that they show why this experiments has not been repeated to anwer the quiestions. Thus I suggest that a justification for it is also included in the main text. This also applies for the qPCR results.
We agree with the reviewer and have added some more explanation for this in the results and also in the discussion. The following sentences are added:
Results: However, considering the low standard deviations for a total of 5 larvae per time point and the sensitivity of the MBL method, we can conclude that M. tuberculosis can survive up to 9 days after infection of zebrafish embryos, but at very low numbers. This is not surprising, considering that a 9 days period of growth is still very short for M.tuberculosis growing at a suboptimal temperature in a hostile environment. However, these low numbers will make any accurate quantitative analysis using follow up MBL experiments very difficult if not impossible. Considering the low numbers of larvae available and the conclusive evidence from the microscopy results we therefore have not replicated the experiment.
Discussion: Although, quantification of the Mtb infection burdens in zebrafish using MBL at the stage up to 9 dpi holds little promise, it might be of use to study later stages of infection in follow up experiments directed at studying the function of the adaptive immune system in defense against tuberculosis in the zebrafish model.
Overall my suggestions is that the authors re-consider the way they present the results they have given the impossibility of repeating some of them to re-inforce their points, and this way solve the flaws in methodology. Simmilarly, the flow of the paper need to be re-consider to avoid misconceptions of their main findings.
We thank the reviewer for this comment. On rereading our paper with this comment in mind we agree that the abstract gives this impression. Therefore we have changed the abstract and the end of the introduction to make more clear what is scientific highlight of the paper.
The end of the abstract now reads:
. The generated extensive transcriptome data sets will be of great use to ad translational value to zebrafish as a model for infection of tuberculosis using the M.marinum infection system. In addition we identify new marker genes such as dusp8 and CD180 that are induced by M.tuberculosis infection in zebrafish and in human macrophages at later stages of infection that can be further investigated.
The last sentence also is now included at the end of the discussion. In addition we have deleted the sentence on the MBL results from the abstract. We hope now it is much clearer what is the purpose of this paper.
Reviewer 4 Report (New Reviewer)
Comments and Suggestions for Authors
That was very helpful and the paper is acceptable in its
new modified form. It is now a cautionary tale lest others want to try
to use zebrafish to study tuberculosis.
/
Author Response
We thank the reviewer for the comments
This manuscript is a resubmission of an earlier submission. The following is a list of the peer review reports and author responses from that submission.
Round 1
Reviewer 1 Report
Comments and Suggestions for Authors
The manuscript “The human pathogen Mycobacterium tuberculosis and the fish 2
pathogen Mycobacterium marinum trigger the same core set of 3 late immune response genes in zebrafish larvae” by Ron P. Dirks1, Anita Ordas, Susanne Jong-Raadsen, Sebastiaan B. Brittijn, Mariëlle C. Haks, Christiaan V. Henkel, Katarina Oravcova, Peter I. Racz, Malgorzata I. Wiweger, Stephen H. Gillespie, Annemarie H. Meijer, Tom H.M. Ottenhoff, Hans J. Jansen and Herman P. Spaink, reports the investigation of Mycobacterium tuberculosis infections in zebrafish. The originality of the reported work relies on the early infection of the zebrafish egg through automatic injection in the yolk of fertilized eggs. After such early infections, the authors report bacterial survival lasting up to 9 days in the developing fish. This work is a real “tour de force” because they have to deal with numerous constraints from the difficulty of injecting class 3 pathogens to the problem of optimal temperature for the experiments with a pathogen having an optimal growth temperature of 37°C and a host with an optimal growth temperature of 28°C. The drawback of the compromise they found is the very small number of fish available for their analysis at up to 9 days post infections despite the very high number of infected eggs. This low number of fish that they could analyze is the reason why some of their results are not statistically robust.
They analyze bacterial loads and transcriptional response in the infected developing zebrafish and compare the transcriptional response to that in Mycobacterium marinum infected zebrafish, and to that in Mycobacterium tuberculosis infected human macrophages. They highlight a core set of commonly induced genes.
Important issues:
While fig1 very clearly states “MtbBerlin” or “MTB DsRed”, and which analysis have been performed on which sets of experiments, the “result” section often refers to M. tuberculosis without further details (see MBL assay lines 270-280). Please be careful to clearly state which bacteria were used.
Some sentences are difficult to understand and shall be rewritten, as for example lines 452-454, 455-456, 466-458, 474-477, 494-496, …. The whole article should be scrutinized.
The authors should discuss the reported differences between the results obtained with “MtbBerlin” and those obtained with “MtbDsRed” and, more specifically stress the importance of using DsRed labelled bacteria to have the possibility of selecting larvae that harbor bacteria prior to measure bacterial load or transcriptional response of the host.
The authors should discuss the fact that IL1b peaks at 5-6 dpf while the infection persists to latter time points.
The authors should clarify whether the embryos from infected batches have been screened microscopically for the presence of bacteria before being processed to measure bacterial loads and/or prepare RNAs for qPCR or transcriptomic analysis.
Minor and/or specific issues :
Line 143: “powdered dry food”, which reference/provider
Lines 147-149: “amikacin treatment”, is before or after paraformaldehyde fixation?
Line 176: Please introduce the “MBL” abbreviation when it first appears in the manuscript.
Line 194: “the corresponding melting temperatures”. Is it “melting” or “annealing”? Shall the reader understand that the melting/annealing temperature is compatible with the polymerization by the enzyme present in the kit?
Line 194: “final melting curve of 81 cycles”. Do the author mean “cycles” or “steps”.
Line 233: “However, general, show no obvious differ”, is there a word missing after “general”?
Line 260: What do the authors mean by “Stereo fluorescent images”?
Lines 270-280: It is not clear whether the embryo/larvae used had been injected with “MtbBerlin” or with “Mtb DsRed”, and if the presence of fluorescent bacteria/granuloma had been scored before RNA isolation.
Lines 288-300: It is not clear for me why the authors have used the “MtbBerlin” injected rather than the “MTB DsRed” injected larvae (do I understand well?). Sorting the larvae based on the presence of fluorescent bacteria would have helped understanding the data and especially the fact that IL1b and mmp9 expression are so low at the latest time point. Same for Illumina deep transcriptome sequencing.
Lines 288-330: It is not clear for me if the same RNA preparations have been used for qPCR and for Illumina sequencing. Please make it clearer.
Figure 1: Shall the reader guess that “MycoPrep/CFU counting” includes MBL assay?
In figure 3, it would be very useful to have dotted lines highlighting the outline of the fish to help understand the fluorescence image and how it corresponds to the appended bright-field image that has another scale.
Figure 6: Please make it clearer, in the figure, that B is “upregulated” and C “downregulated” so that the figure would become self-explanatory.
Figure 6 legend appears twice (lines 332-338 and 342-348)
Comments on the Quality of English Language
Some sentences are difficult to understand and shall be rewritten, as for example lines 452-454, 455-456, 466-458, 474-477, 494-496, …. The whole article should be scrutinized.
Author Response
Reviewer 1:
The manuscript “The human pathogen Mycobacterium tuberculosis and the fish 2
pathogen Mycobacterium marinum trigger the same core set of 3 late immune response genes in zebrafish larvae” by Ron P. Dirks1, Anita Ordas, Susanne Jong-Raadsen, Sebastiaan B. Brittijn, Mariëlle C. Haks, Christiaan V. Henkel, Katarina Oravcova, Peter I. Racz, Malgorzata I. Wiweger, Stephen H. Gillespie, Annemarie H. Meijer, Tom H.M. Ottenhoff, Hans J. Jansen and Herman P. Spaink,
reports the investigation of Mycobacterium tuberculosis infections in zebrafish. The originality of the reported work relies on the early infection of the zebrafish egg through automatic injection in the yolk of fertilized eggs. After such early infections, the authors report bacterial survival lasting up to 9 days in the developing fish. This work is a real “tour de force” because they have to deal with numerous constraints from the difficulty of injecting class 3 pathogens to the problem of optimal temperature for the experiments with a pathogen having an optimal growth temperature of 37°C and a host with an optimal growth temperature of 28°C. The drawback of the compromise they found is the very small number of fish available for their analysis at up to 9 days post infections despite the very high number of infected eggs. This low number of fish that they could analyze is the reason why some of their results are not statistically robust.
They analyze bacterial loads and transcriptional response in the infected developing zebrafish and compare the transcriptional response to that in Mycobacterium marinum infected zebrafish, and to that in Mycobacterium tuberculosis infected human macrophages. They highlight a core set of commonly induced genes.
Reply: We thank the reviewer for the understanding of the technical difficulty and relevance of this work and the many useful comments.
Important issues:
While fig1 very clearly states “MtbBerlin” or “MTB DsRed”, and which analysis have been performed on which sets of experiments, the “result” section often refers to M. tuberculosis without further details (see MBL assay lines 270-280). Please be careful to clearly state which bacteria were used.
Response: we have now indicated also when a strain has no plasmid. “(no plasmid)” in the indicated lines and also in the legend.
Some sentences are difficult to understand and shall be rewritten, as for example lines 452-454, 455-456, 466-458, 474-477, 494-496, …. The whole article should be scrutinized.
Response: We have made these sentences more clear.
The authors should discuss the reported differences between the results obtained with “MtbBerlin” and those obtained with “MtbDsRed” and, more specifically stress the importance of using DsRed labelled bacteria to have the possibility of selecting larvae that harbor bacteria prior to measure bacterial load or transcriptional response of the host.
Response: we have made more clear that we have not used dsRed labelled bacteria for transcriptional studies. The reason this is not used is because at high safety levels we have no means to use fluorescence as a marker for selecting infected larvae prior to RNA isolation. We do not see any indications that the dsRed strain is different than the wild type strain in the survival curves (although there are no sufficient numbers of replicates to test significance). This is also not expected based on the published literature which shows no effect of fluorescent reporter genes on virulence of mycobacteria in animal models.
The authors should discuss the fact that IL1b peaks at 5-6 dpf while the infection persists to latter time points.
Response: We have now mentioned this explicitly in the discussion section.
The authors should clarify whether the embryos from infected batches have been screened microscopically for the presence of bacteria before being processed to measure bacterial loads and/or prepare RNAs for qPCR or transcriptomic analysis.
Response: we have now made this more clear in the text and the legends.
Minor and/or specific issues :
Line 143: “powdered dry food”, which reference/provider
Response: We have now supplied these details
Lines 147-149: “amikacin treatment”, is before or after paraformaldehyde fixation?
Response: This was done after fixation, we have now made this clear
Line 176: Please introduce the “MBL” abbreviation when it first appears in the manuscript.
Response: We have now mentioned this at the first appearance.
Line 194: “the corresponding melting temperatures”. Is it “melting” or “annealing”? Shall the reader understand that the melting/annealing temperature is compatible with the polymerization by the enzyme present in the kit?
Response: “Annealing” is correct and we have changed this in the text.
Line 194: “final melting curve of 81 cycles”. Do the author mean “cycles” or “steps”.
Response: “Steps” is correct and we have changed this in the text.
Line 233: “However, general, show no obvious differ”, is there a word missing after “general”?
Response: This error is corrected
Line 260: What do the authors mean by “Stereo fluorescent images”?
Response: we have now changed the wording to : Images taken with a fluorescent stereomicroscope
Lines 270-280: It is not clear whether the embryo/larvae used had been injected with “MtbBerlin” or with “Mtb DsRed”, and if the presence of fluorescent bacteria/granuloma had been scored before RNA isolation.
Response: This now made clear, see response above
Lines 288-300: It is not clear for me why the authors have used the “MtbBerlin” injected rather than the “MTB DsRed” injected larvae (do I understand well?). Sorting the larvae based on the presence of fluorescent bacteria would have helped understanding the data and especially the fact that IL1b and mmp9 expression are so low at the latest time point. Same for Illumina deep transcriptome
Response: see response above. The reason this is not used is because at high safety levels we have no means to use fluorescence as a marker for selecting infected larvae prior to RNA isolation. The fixation protocol to get the larvae out of the high safety lab is not compatible with the RNA isolation protocol.
Lines 288-330: It is not clear for me if the same RNA preparations have been used for qPCR and for Illumina sequencing. Please make it clearer.
Response:we have now mentioned that these samples are the same as used for the RNAseq experiments.
Figure 1: Shall the reader guess that “MycoPrep/CFU counting” includes MBL assay?
Response: We have now expanded the legend for Figure 1 to make this clear.
In figure 3, it would be very useful to have dotted lines highlighting the outline of the fish to help understand the fluorescence image and how it corresponds to the appended bright-field image that has another scale.
Response: we have added scale bars to the figure to make this more clear.
Figure 6: Please make it clearer, in the figure, that B is “upregulated” and C “downregulated” so that the figure would become self-explanatory.
Response: we have now added this text to the figure to make it more clear.
Figure 6 legend appears twice (lines 332-338 and 342-348)
Response: this error was introduced by the editorial reformatting. This will be corrected.
Comments on the Quality of English Language
Some sentences are difficult to understand and shall be rewritten, as for example lines 452-454, 455-456, 466-458, 474-477, 494-496, …. The whole article should be scrutinized.
Response:We have corrected several textual errors and reformulated some sentences.
Reviewer 2 Report
Comments and Suggestions for Authors
1- Authors should substitute the keywords that are present in the title (keywords are words representative of the research and not present in the title).
2- Report 2023 WHO; available from: http://www.who.int/tb/publications/global_report/en/. It is better to be added as a reference (WHO 2023).
3- The conclusion should be deduced in one paragraph.

Author Response
- Authors should substitute the keywords that are present in the title (keywords are words representative of the research and not present in the title).
Response: we have removed these key words and added some additional key words
- Report 2023 WHO; available from: http://www.who.int/tb/publications/global_report/en/. It is better to be added as a reference (WHO 2023).
Response: We have now made this into a reference
3- The conclusion should be deduced in one paragraph.
Response: We have shortened the conclusion section with two sentences.
Reviewer 3 Report
Comments and Suggestions for Authors
1. I appreciate the opportunity to review the manuscript titled "The human pathogen Mycobacterium tuberculosis and the fish pathogen Mycobacterium marinum trigger the same core set of late immune response genes in zebrafish larvae" for potential publication in the Biology. However, there are some concerns that need to be addressed in order to consider it for publication.
2. The references cited in this manuscript provide support for the statements made. However, it is recommended to include additional recent references to ensure the information remains up to date. Since, the manuscript has been cited in 1in 8.69% (2/23) of the most recent publications within the past five years., including 1 publication in 2020, and 1 publication in 2023.
3. Introduction section presents and justifies the purpose of research in very general terms. The Introduction should outline the shortest logical path through the critical literature that can support the study's hypotheses and/or aims.
4. I suggest that the author consider removing Table 1, which compares M. tuberculosis and M. marinum, from the introduction.
5. I recommend adding a subsection dedicated to "Statistical Analysis" within the Method section. This addition will provide readers with clear insight into the statistical methods and tests employed, enhancing the transparency and reproducibility of the study's findings.
6. Employing various colors in the Kaplan-Meier plot for zebrafish larvae survival analysis can greatly improve clarity and visual interpretation. I also believe that further enhancing the analysis of survival rate and time through statistical methods would significantly contribute to the robustness of the findings. Additionally, employing the log-rank test for analyzing the Kaplan-Meier plot could further strengthen the methodology and provide valuable insights.
7. It is advisable to include a scale bar in Figure 3, which depicts the fluorescent microscopy imaging of Mycobacterium tuberculosis infection in zebrafish larvae. The addition of a scale bar will provide essential spatial reference and aid in accurately assessing the size and dimensions of the observed features, thereby enhancing the interpretability of the microscopy images.
8. It is essential to incorporate error bars into Figure 4 and Figure 6A.
9. Presenting the CFU data using log values would provide a more accurate depiction of the bacterial load variation and enhance the visual interpretation of the results. This approach enables a better grasp of the exponential nature of bacterial growth or decline and establishes a standardized method for reporting microbiological data.
10. In the context of statistical analysis of the qPCR results presented in Supplementary Figure 1, the presence of board error bars is unexpected and has the potential to influence the interpretation of the data. Error bars are essential for comprehending data variation and uncertainty, as they offer valuable insights into measurement precision and reliability. Furthermore, the omission of a normality test before conducting the Mann-Whitney U test raises concerns regarding the validity of the statistical analysis. It is imperative to address these issues to uphold the accuracy and dependability of the study's findings.
11. It's essential to specify the units or scale of the Y-axis in Figure 5, which depicts the differential gene expression of IL1b and mmp9.
12. To accurately represent the gene, "IL1b" should be replaced with "IL-1β" in the context of the differential gene expression data. This adjustment ensures precision and consistency in the nomenclature of the gene, aligning with standard naming conventions.
13. I would recommend reviewing the entire article to ensure that gene symbols are properly italicized, while symbols for proteins are not. This will help maintain consistency and accuracy in the representation of genetic elements and proteins throughout the article, aligning with standard conventions.
14. I suggest rewriting the 2nd and 3rd paragraphs in the discussion section to avoid repetition of the results and instead focus on introducing meaningful discussion points supported by relevant references. By doing so, the discussion will be enriched with critical analysis and contextualization, enhancing the overall depth and insight of the manuscript. This revision will ensure that the discussion section effectively contributes to the scholarly discourse on the topic at hand.
15. I would recommend that the author thoroughly addresses the study's limitations within the discussion section. By acknowledging and discussing the limitations, the author can provide a more comprehensive and transparent interpretation of the findings, enhancing the overall quality of the manuscript.
16. I recommend that the author concisely states the primary findings and practical applications in the conclusion to enhance the manuscript's impact and readability. This will effectively guide the reader's understanding of the research outcomes and their potential significance in the field.
17. Please ensure that for writing research articles, the full name is written the first time followed by the abbreviation in parentheses (e.g., PVP, PBS, BSL). Kindly check throughout the article for consistency.
Author Response
- I appreciate the opportunity to review the manuscript titled "The human pathogen Mycobacterium tuberculosis and the fish pathogen Mycobacterium marinum trigger the same core set of late immune response genes in zebrafish larvae" for potential publication in the Biology. However, there are some concerns that need to be addressed in order to consider it for publication.
- The references cited in this manuscript provide support for the statements made. However, it is recommended to include additional recent references to ensure the information remains up to date. Since, the manuscript has been cited in 1in 8.69% (2/23) of the most recent publications within the past five years., including 1 publication in 2020, and 1 publication in 2023.
Response: we have added a more recent review reference of 2023.
- Introduction section presents and justifies the purpose of research in very general terms. The Introduction should outline the shortest logical path through the critical literature that can support the study's hypotheses and/or aims.
Response: We thanks the reviewer for the comments.
- I suggest that the author consider removing Table 1, which compares M. tuberculosisand M. marinum, from the introduction.
We believe that this table make the paper more accessible to a general public that is not used to using M.marinum as a model for tuberculosis. We have added that M.marinum also give rise to zoonosis to make the table more informative.
- I recommend adding a subsection dedicated to "Statistical Analysis" within the Method section. This addition will provide readers with clear insight into the statistical methods and tests employed, enhancing the transparency and reproducibility of the study's findings.
Response: we now have added a section for statistical analysis. To make the data analysis totally transparent we have now also provided supplementary tables with the raw data and the statistical analyses.
- Employing various colors in the Kaplan-Meier plot for zebrafish larvae survival analysis can greatly improve clarity and visual interpretation. I also believe that further enhancing the analysis of survival rate and time through statistical methods would significantly contribute to the robustness of the findings. Additionally, employing the log-rank test for analyzing the Kaplan-Meier plot could further strengthen the methodology and provide valuable insights.
Response: We have now included a color version of this figure.
- It is advisable to include a scale bar in Figure 3, which depicts the fluorescent microscopy imaging of Mycobacterium tuberculosis infection in zebrafish larvae. The addition of a scale bar will provide essential spatial reference and aid in accurately assessing the size and dimensions of the observed features, thereby enhancing the interpretability of the microscopy images.
- It is essential to incorporate error bars into Figure 4 and Figure 6A.
Response: Figure 4 is based on only one biological replicate. To include error bars for the technical replicates could be misleadingly suggest that there have been biological replicates. However, to make the data and analysis completely transparent we have added a supplementary table with all the raw data, standard deviations and the derivative data.
In Fig. 6A: these are exact numbers and are therefore without a statistical error. It represents the numerical outcome of analyses of which the P values are provided.
- Presenting the CFU data using log values would provide a more accurate depiction of the bacterial load variation and enhance the visual interpretation of the results. This approach enables a better grasp of the exponential nature of bacterial growth or decline and establishes a standardized method for reporting microbiological data.
Response: We have a supplementary table with all the raw data, standard deviations and the derivative data. Readers are now able to simply output the data in any possible representation. Considering the small variations providing a log scale would not be our preference.
- In the context of statistical analysis of the qPCR results presented in Supplementary Figure 1, the presence of board error bars is unexpected and has the potential to influence the interpretation of the data. Error bars are essential for comprehending data variation and uncertainty, as they offer valuable insights into measurement precision and reliability. Furthermore, the omission of a normality test before conducting the Mann-Whitney U test raises concerns regarding the validity of the statistical analysis. It is imperative to address these issues to uphold the accuracy and dependability of the study's findings.
Responses: We appreciate the comments on the statistics of this analysis. The large variation as shown by the broad error bars is due to the relatively small sample size and the inherent variation of the survival of the larvae. We value reporting these variations as shown by the error bars, for transparent interpretation of the variation in the sample.
We performed the normality test as standard within this analysis, but omitted it from the manuscript. It is now mentioned that the Q-PCR scored poorly for normality. This is now also explicitly mentioned in the Methods (comment 5). Given the small numbers, non-normality is not surprising, and the Mann-Whitney test (which assumes non-normality) was statistically significant.
Markers IL1B and MMP9 were chosen because they are very robust in zebrafish inflammation assays. As can be seen in the raw data we have hardly any variation in the background levels in the absence of infection and therefore is in perfect agreement with published data from zebrafish studies using these markers. We have now added a sentence in the manuscript to stress this point.
- It's essential to specify the units or scale of the Y-axis in Figure 5, which depicts the differential gene expression of IL1b and mmp9.
Response: This has now been added for figure 5 and also for supplementary Figure1.
- To accurately represent the gene, "IL1b" should be replaced with "IL-1β" in the context of the differential gene expression data. This adjustment ensures precision and consistency in the nomenclature of the gene, aligning with standard naming conventions.
Response: In zebrafish approved nomenclature (ZFIN) il1b is the correct gene name. We have now avoided everywhere to use the protein name.
https://zfin.org/ZDB-GENE-040702-2#summary
- I would recommend reviewing the entire article to ensure that gene symbols are properly italicized, while symbols for proteins are not. This will help maintain consistency and accuracy in the representation of genetic elements and proteins throughout the article, aligning with standard conventions.
Response: We have corrected this in various instances also in the tables.
- I suggest rewriting the 2nd and 3rd paragraphs in the discussion section to avoid repetition of the results and instead focus on introducing meaningful discussion points supported by relevant references. By doing so, the discussion will be enriched with critical analysis and contextualization, enhancing the overall depth and insight of the manuscript. This revision will ensure that the discussion section effectively contributes to the scholarly discourse on the topic at hand.
Response: we agree that there was unneeded repetition in these paragraphs that has now been removed. We have replaced this by more in depth discussion.
- I would recommend that the author thoroughly addresses the study's limitations within the discussion section. By acknowledging and discussing the limitations, the author can provide a more comprehensive and transparent interpretation of the findings, enhancing the overall quality of the manuscript.
Response: we have added text on the limitation of zebrafish to predict translational value based on homologies of genes and presented some additional references. We believe we highlighted the limitations of the study.
- I recommend that the author concisely states the primary findings and practical applications in the conclusion to enhance the manuscript's impact and readability. This will effectively guide the reader's understanding of the research outcomes and their potential significance in the field.
Response: we have made the conclusion section more concise.
- Please ensure that for writing research articles, the full name is written the first time followed by the abbreviation in parentheses (e.g., PVP, PBS, BSL). Kindly check throughout the article for consistency.
Response: we have now checked this through the entire manuscript and revised accordingly.
Round 2
Reviewer 3 Report
Comments and Suggestions for Authors#3 I would recommend that the author revisits the last paragraph of the Introduction to provide a clearer and more specific outline of the purpose of the research. This adjustment will further enhance the coherence and logical flow of the Introduction, aligning it more closely with the critical literature that supports the study's hypotheses and aims. Thank you for your attention to this matter.
#6 While I appreciate the inclusion of a color version of the Kaplan-Meier plot, I would like to inquire about the incorporation of further analysis of survival rate and time through statistical methods, as well as the consideration of employing the log-rank test for analyzing the Kaplan-Meier plot. These additions would significantly strengthen the methodology and provide valuable insights into the robustness of the findings. I would kindly request the author's attention to these aspects.
#8 I appreciate the effort to provide transparency by including the raw data and standard deviations in a supplementary table. However, given the importance of biological replicates for reliable conclusions, I encourage exploring the possibility of repeating the experiment to further enhance the robustness of the findings. If there are specific obstacles preventing the repetition of the experiment, it would be valuable to provide a clear explanation in the manuscript.
#9 I would like to reiterate the importance of using log colony-forming units (CFU) values to compare or show changes in bacterial load. This approach provides a more accurate representation of changes and aligns with standard practices for microbiological data analysis. Utilizing log CFU values not only facilitates visual interpretation but also supports the stability of variance and the applicability of statistical tests. Thank you for considering this recommendation.
Author Response
We thank the reviewer for the additional suggestions. Below a point by point response.
#3 I would recommend that the author revisits the last paragraph of the Introduction to provide a clearer and more specific outline of the purpose of the research. This adjustment will further enhance the coherence and logical flow of the Introduction, aligning it more closely with the critical literature that supports the study's hypotheses and aims. Thank you for your attention to this matter.
Response:
We have changed the end of the introduction slightly to more show the purpose of this study.
#6 While I appreciate the inclusion of a color version of the Kaplan-Meier plot, I would like to inquire about the incorporation of further analysis of survival rate and time through statistical methods, as well as the consideration of employing the log-rank test for analyzing the Kaplan-Meier plot. These additions would significantly strengthen the methodology and provide valuable insights into the robustness of the findings. I would kindly request the author's attention to these aspects.
Response:
We have now included the results of log-rank tests in an additional supplementary figure. For total transparency we have now also added a supplementary file with the raw data.
#8 I appreciate the effort to provide transparency by including the raw data and standard deviations in a supplementary table. However, given the importance of biological replicates for reliable conclusions, I encourage exploring the possibility of repeating the experiment to further enhance the robustness of the findings. If there are specific obstacles preventing the repetition of the experiment, it would be valuable to provide a clear explanation in the manuscript.
Response: we are not able to repeat any of the experiments. We would like to stress that experiments with TB in a high safety BSL3 lab and with ethical permits for zebrafish for extended study are extremely difficult.
#9 I would like to reiterate the importance of using log colony-forming units (CFU) values to compare or show changes in bacterial load. This approach provides a more accurate representation of changes and aligns with standard practices for microbiological data analysis. Utilizing log CFU values not only facilitates visual interpretation but also supports the stability of variance and the applicability of statistical tests. Thank you for considering this recommendation.
Response: we have now added an additional supplementary figure with log CFU representation.